# Sub3DNet1.0: A deep learning model for regional-scale 3D subsurface structure mapping

Zhenjiao Jiang [1,2], Dirk Mallants [2], Lei Gao [2], Tim Munday [3], Gregoire Mariethoz [4], Luk Peeters [3]

1 Key Laboratory of Groundwater Resources and Environment, Ministry of Education, College of Environment and Resources, Jilin University, Changchun, 130021, China

2 CSIRO Land & Water, Locked Bag 2, Glen Osmond, SA 5064, Australia

3 CSIRO Mineral Resources, Locked Bag 2, Glen Osmond, SA 5064, Australia

4 University of Lausanne, Faculty of Geosciences and Environment, Institute of Earth Surface Dynamics, Lausanne, Switzerland

*Corresponding author:* Zhenjiao Jiang (jiangzhenjiao@hotmail.com)

**Abstract.** This study introduces an efficient deep learning model based on convolutional neural networks with joint autoencoder and adversarial structures for 3D subsurface mapping from 2D surface observations. The method was applied to delineate palaeovalleys in an Australian desert landscape. The neural network was trained on a 6,400 km$^2$ domain by using a land surface topography as 2D input and an airborne electromagnetic (AEM)-derived probability map of palaeovalley presence as 3D output. The trained neural network has a squared error < 0.10 across 99% of the training domain and produces a squared error < 0.10 across 93% of the validation domain, demonstrating that it is reliable in reconstructing 3D palaeovalley patterns beyond the training area. Due to its generic structure, the neural network structure designed in this study and the training algorithm have broad application potential to construct 3D geological features (e.g. ore bodies, aquifer) from 2D land surface observations.

## 1 Introduction

Imaging the Earth's subsurface is crucial for the exploration and management of mineral, energy and groundwater resources, its reliability depending on the availability and quality of geological data. Although the amount and quality of geological data obtained from borehole logs, geophysical prospecting and remote sensing has increased over the past decades, their spatial distribution is highly uneven. Big data sets on geology and geomorphology are globally available either as land surface observations (typically remote sensing and topographical data and their derivatives), or only regionally available in a limited number of highly-developed mining and oil fields (e.g., downhole, surface and airborne geophysical data and interpretations). In Australia, as an example, the former are readily available at relatively low, or no-cost, while the latter are often non-existing and expensive for remote desert areas where a key challenge is to secure groundwater for town/community supply, primarily from shallow aquifers (Munday, 2020a, b). In their study, Munday et al. (2020a, b) interpreted 17,000 line km of airborne electromagnetic (AEM) data covering an area of about 30,000 km$^2$ within the much larger Great Victoria Desert in central Australia (422,000 km$^2$). With a regional AEM line spacing of 2 km, smaller infill areas were defined close to remote isolated communities where line spacing was reduced to 250 and 500 m. This provided greater detail of the character of the subsurface

electrical conductivity, enabling more accurate mapping of palaeovalley aquifers to be achieved (Munday, 2020a). Unfortunately the application of such high-resolution data to much larger areas like the entire Victorian Desert would be cost prohibitive, so alternative approaches to the definition of palaeovalley systems are required.

Commonly used methods for modelling complex geological structures include geostatistical approaches such as sequential Gaussian or indicator simulation (Lee et al., 2007), transition probability simulation (Felletti et al., 2006; Weissmann and Fogg, 1999), or multiple-point simulation (MPS) methods (Hu and Chugunova, 2008; Mariethoz and Caers, 2014; Strebelle, 2002). Most geostatistical approaches are suitable for "interpolation", which performs well in predicting 3D subsurface structures within the data-rich region (Kitanidis, 1997). However, their ability to "extrapolate" a 3D subsurface structure is limited. Alternatively, MPS is an advanced method to quantify the complex spatial structure based on training images. It transfers the quantified structures to the data-scarce region for stochastic predictions; however, a realistic 3D training image is difficult to obtain. Overall, most existing subsurface structure modelling approaches are developed to analyse a single-support dataset, that is, the data types employed to define spatial structures are presumed to be identical to as those employed for predictive purposes (de Marsily et al., 2005). Better defining and using the relationships between multiple-support datasets allows regional-scale subsurface structural imaging based on easy-to-obtain, lower costs data sets. However, existing methods are often ineffective or inefficient in capturing essential features and patterns from available large and multiple-support datasets. The analysis of multiple support datasets, e.g. downhole geophysical logs and 3D reflection seismic transects with lithofacies, is still based on subjective expert knowledge. A fast and reliable tool capable of deriving a robust relationship among multiple-support big datasets is needed for improved high-resolution imaging of 3D subsurface structures.

Deep learning approaches specialised in big data mining have the potential to fill this gap (Gu et al., 2018; Hinton and Salakhutdinov, 2006; Marcais and de Dreuzy, 2017). Applications in the geosciences include earthquake detection based on seismic monitoring (Mousavi and Beroza, 2019; Perol et al., 2018), and disaster recognition from remote sensing data (Amit et al., 2016; Längkvist et al., 2016), among others. Complex subsurface geological structures, such as palaeovalley fill boundaries and thickness, have also been predicted based on a digital elevation model combined with deep learning, assuming that the an extension of the palaeovalley topography is still present at land surface (Mey et al., 2015). A recent breakthrough in deep learning are the 2D to 3D image processing approaches (Niu et al., 2018; Sinha et al., 2017; Wu et al., 2016; Yi et al., 2017). Such approaches give confidence that novel ways to rapidly and automatically identify buried 3D subsurface structures directly from readily-available 2D surface observations (e.g. digital elevation models, land cover maps, and signals captured by airborne geophysical surveys) is feasible. This is most obvious where the 3D subsurface structures have a relationship with the 2D surface observations, even though this relationship may be obscured or even unknown. A neural network framework that reliably transforms 2D input data into 3D output data is required that has the flexibility to fuse multiple types of geology and geophysical input data for more complex 3D geological subsurface structure imaging.

To this end, we designed a deep convolutional neural network (CNN) with joint autoencoder (Kingma and Welling, 2013) and adversarial structures (Goodfellow et al., 2014). The autoencoder component features large input and output images connected by a small latent space. This structure is advantageous for the fusion of complex input data and 3D image reconstruction. Its training involves direct back-propagation according to a voxel-wise independent heuristic criterion, and thus often needs a large training dataset to constrain the model and avoid overfitting (Laloy et al., 2018). The generative adversarial learning is capable to generate multiple images inheriting the probability structure of one real image, which can relax the need for very large training dataset.

To demonstrate that the interplay between autoencoder and adversarial components is capable of effectively exploiting land surface data to generate regional-scale buried 3D geological structures, the proposed neural network model is applied to an Australian desert landscape to generate regional-scale 3D palaeovalley patterns from 2D digital terrain information. The case study area is a pre-Pliocene palaeovalley system in central Australia that has been postulated to contain significant groundwater resources (Dodds and Sampson, 2000). In these very old landscapes the definition of the palaeovalley systems has, until more recently, remained relatively poorly known (Munday, 2020a), which is attributed to aeolian, alluvial and colluvial sediments forming a continuous cover over much of the region to depths exceeding 50m. Below this depth the definition of the palaeovalley systems becomes significantly clearer with a well-defined network of major alluvial channels and tributary systems, evident from the analysis of AEM images (Munday, 2020a). However, large parts of the region have no geophysical data coverage of value for defining near surface aquifer systems, and therefore developing predictive techniques to help target ground investigations can be of significant value.

Our goal is therefore to employ the proposed model to express the relationship between an easy-to-obtain dataset (readily available landsurface information) and a more costly dataset (AEM-derived palaeovalley pattern) for the specific purpose of detecting palaeovalley features that would facilitate the discovery of new groundwater resources in arid and semi-arid regions. Specifically, the primary aim of using geophysical methods combined with topographical data is to define the form and nature of palaeovalleys to assist with siting groundwater boreholes in the deepest part of the palaeovalley. The model uses AEM only for model development on a small training area while the application (i.e. detection of palaeovalleys across large areas) uses readily available landsurface information that otherwise (i.e., without AEM coupling through a training procedure) would have had potentially little value for palaeovalley detection. Such methodology is premised on the existence of a mechanistic or empirical connection between landsurface features and subsurface distribution of palaeovalleys. To what degree such correlation exists (and can be cast in a predictive framework) between palaeovalley geometry and landsurface features derived from digital elevation data in the palaeovalley system of the Musgrave Province can be determined using a deep convolutional neural network methodology. It is worthy of note that the topography of the study area is very subdued, and whilst the contemporary draining channels are discordant with respect to their ancient precursors as defined by the thalweg or deepest part of the palaeovalley, these old valley systems are concordant with the subdued valley forms expressed in today's landscape.

## 2 Background, Materials and Method

### 2.1 Genesis of palaeovalley systems in central Australia

The genesis of the central Australia palaeovalleys of the Musgrave Province (including the Great Victorian Desert and the APY Lands as our study area) covers about 60 Ma, and started as early as the Mid-Late Mesozoic to Early Palaeogene (about 65 Ma) with the latest palaeovalley infilling completed during the Early to Late Pliocene (about 2.5-5 Ma ago). The palaeovalley history involves a sequence of fluvial depositional periods interrupted by marine incursions, with climatic boundary conditions ranging from warm and humid (Late Miocene) to more aridic conditions (Late Pliocene to Early Pleistocene) (Krapf et al., 2019).

Valley incision was preceded by deep weathering of exposed basement rocks in the Mid to Late Mesozoic (Alley et al., 2009). While timing of the incision is debated, Hou et al. (2008) considered that the first infilling of the palaeovalleys with sandy fluvial deposits occurred through the Late Mesozoic – Early Palaeogene (about 65 Ma ago) and was focused along long lived, and still active (Pawley et al., 2014), structural discontinuities within the faulted Mesoproterozoic crystalline basement rocks (Figure 1). In the subsequent Late Miocene to Early Pliocene (about 40-13 Ma), characterized by a warm and humid climate, both freshwater and marine environment reversals occurred with marine sediments being deposited, transitioning to brackish and freshwater lakes (playas) occupying the valley floor. During the Late Miocene to Early Pliocene (about 10-3 Ma ago), evaporation of these sediments led to the deposition of a gypsum layer which was accompanied by intermittent fluvial deposition. A combination of active faulting and sedimentation may have encouraged the development of small, narrow internally draining basins during this period The second and final fluvial deposition with quartz-rich sands occurred during the wetter Early to Late Pliocene (about 2.5-5 Ma ago). After this, the sedimentation continued into the Quaternary, with deposition of fluvial and colluvial sediments across the aridic landscape. During the Pliocene – Holocene (about 4 Ma ago to present), sand plains and sand sheets developed as a result of aeolian processes (Krapf et al., 2019; Munday, 2020b).

As a result of this long history of land-forming processes, the valley structures of our study area are complex, with varying width and geometry (Krapf et al., 2019; Munday, 2020b). Whilst fluvial systems at the coarse spatial scale are "continuous", at a finer scale they may be discontinuous – shifting braided channel systems resulting in pinching out of fine or coarse scale sedimentary packages, etc.(Krapf et al., 2019). A not insignificant role in the creation of lateral discontinuities was played by Neotectonics resulting from the reactivation of basement structures, which in the context of the APY Lands (our study site), created discontinuities in both sedimentation and valley development, and important to groundwater systems, formed hydraulic barriers in the overlying sediments. Such variations in width and depth of palaeovalleys can cause discontinuities when airborne electromagnetics are geophysically inverted, particularly if the valleys are smaller than the footprint (resolution) of the airborne system. However, most prominent are discontinuities in the lateral continuation of the conductivity features associated with the valley fill, particularly where major fault systems cross cut the primary orientation of the valley systems.

These become particularly apparent at depth (>50m) in the subsurface (see Munday, 2020b). This is attributed to the effects of active tectonics during the valley fill events.

More important for the success of AEM in deriving palaeovalley features is the variation in the petrophysical properties of the valley fill materials. If those properties vary, then one can expect to see an airborne system varying in its capability to map continuity. The critical factor for deep learning (DL) applications is understanding what the DL-based fit is actually working

The influence of neotectonism on the observed conductivity structure associated with palaeovalley fill sequences has been discussed elsewhere by Munday et al.(2001), while Munday et al. (2016) highlighted the role neotectonics may have played in influencing the patterns of the observed electrical conductivity structure in the Musgrave province of South Australia. These

studies demonstrated the role faults, interpreted in the regional magnetics, play in influencing the presence of abrupt discontinuities in the modelled conductivity structure.

More important for the success of AEM in deriving palaeovalley features is the variation in the petrophysical properties of the valley fill materials. If those properties vary, then one can expect to see an airborne system varying in its capability to map continuity. The critical factor for deep learning (DL) applications is understanding what the DL-based fit is actually working

with. That would determine whether one is fitting a geophysical expression of a geological system, which by its nature will be a simplification of true geology, or geological reality. The geophysical expression is well matched with the geological reality in that targeted drilling (described inKrapf et al., 2019; Munday, 2020b), confirmed the presence of thick (>150m) alluvial sediment fill sequences associated with the interpreted palaeovalleys, which were also coincident with the more conductive linear features identified in the AEM data. For the larger palaeovalleys, where the conductive structures identified correlate

well with alluvial fill of the valley systems, the geophysical data maps geology well. Consequently it is reasonable to argue that DL is fitting geological reality, but at the finer scales a mismatch may occur between geological reality and its geophysical expression. For both scales, however, the DL application will be affected by the underlying geophysical expression of the geology and the inversion approach used. In the latter case we employ a 1D Layered Earth Inversion (LEI) routine with lateral constraints. 1D assumptions in the inversion include the assumption that the earth consists of uniform, laterally extensive

layers. At the scale of AEM system footprint and mapping scale of this study this assumption holds. Similar inversion assumptions have been successfully employed in other studies of palaeovalley systems using 1D inversion codes (e.g. Høyer et al., 2015; Korus et al., 2017). Davis et al. (2016) and Roach et al. (2014) reported on the successful application of 1D inversion approaches with AEM data for delimiting palaeovalley systems in Australian settings.

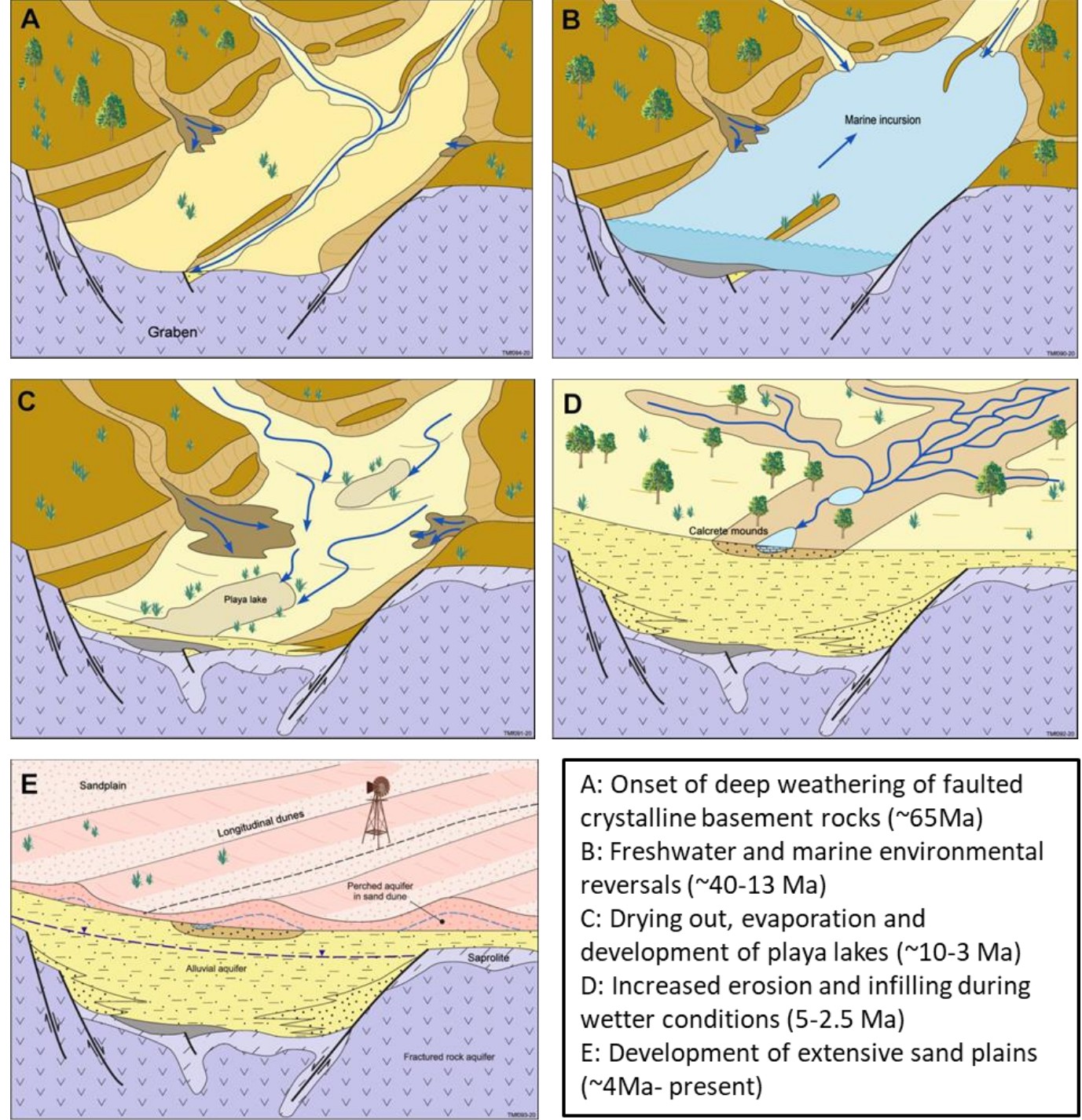

A: Onset of deep weathering of faulted crystalline basement rocks (~65Ma)
B: Freshwater and marine environmental reversals (~40-13 Ma)
C: Drying out, evaporation and development of playa lakes (~10-3 Ma)
D: Increased erosion and infilling during wetter conditions (5-2.5 Ma)
E: Development of extensive sand plains (~4Ma- present)

**Figure 1. Conceptualised genesis of the palaeovalley landscape in the Musgrave Province in South Australia (after Krapf et al., 2019; Munday, 2020b).**

## 2.2 Neural network methodology

The adversarial neural network for 3D subsurface imaging involves three steps: (1) patch extraction and representation, (2) nonlinear mapping and reconstruction, and (3) statistical expression of the generated image (Fig. 2). The first step is referred to as 'encoder' (Fig. 2a), which is employed to fuse the information contained in the 2D land surface observation images (input data) into a low-dimension layer by successive convolutions (Hinton and Salakhutdinov, 2006):

$$h(\mathbf{x}) = f(\mathbf{W} \cdot \mathbf{x} + \boldsymbol{b}), \tag{1}$$

where $f$ is a nonlinear function referred to as "activation function", $\mathbf{W}$ is a matrix of weights and $\boldsymbol{b}$ is a bias vector in the encoder.

The encoder can be designed to contain multiple layers, where the number of layers is defined as 'depth'. Each layer can contain multiple images, with the number of images defined as 'width'. The images in one layer are convoluted to generate the elements in the image of the next layer by weight filters, and the elements in the low-dimension layer of the encoder (the output) are called 'code'. The process of convolution is illustrated in Fig. 2b, which shows that with a filter size of 2×2 (for a 2D image convolution for example), one element in the output layer is related to 4 elements in the input layer. Thus, the spatial correlation scale addressed by the convolutional neural network can be controlled by the filter size in both vertical and horizontal directions.

The weight and bias in the encoder are trained to ensure that the code follows a standard normal distribution, by minimizing the Kullback–Leibler divergence ($L1$), defined as (Kullback and Leibler, 1951):

$$L1 = \frac{1}{2N} \sum_{i=1}^{N} (\mu^2 + \sigma^2 - \log\sigma^2 - 1)_i, \tag{2}$$

where $N$ is number of codes in the final output layer of the encoder, $\mu$ and $\sigma$ are the mean and standard deviation of the codes, respectively.

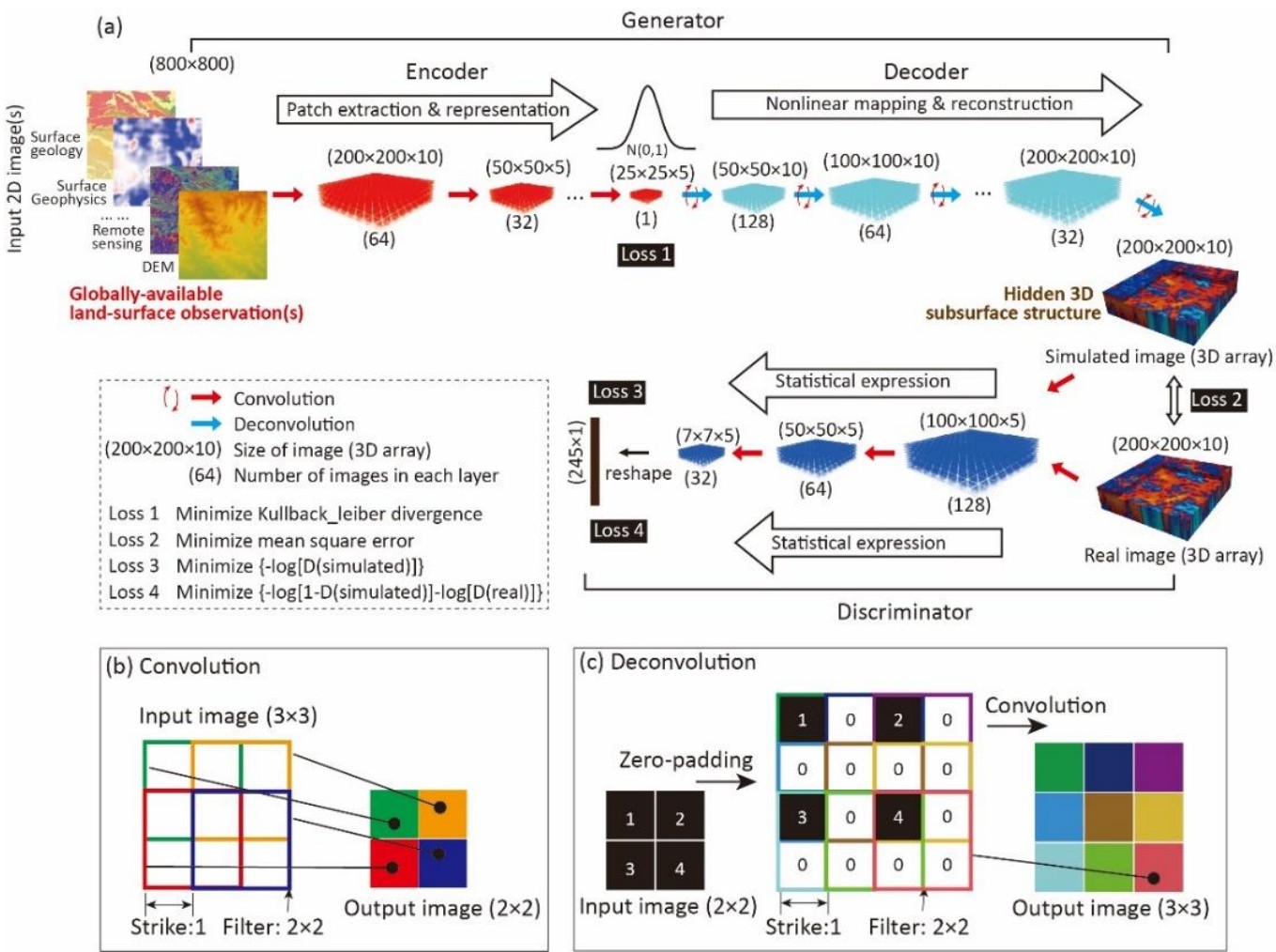

**Figure 2. (a) Adversarial convolution neural network composed of (1) encoder for input image(s) (e.g. surface geology map, uninterpreted geophysics prospecting, remote sensing images and digital elevation model) patch extraction and representation, (2) decoder for nonlinear mapping and 3D image (i.e. hidden subsurface structure) reconstruction, (3)**
**discriminator for distinguishing the output 3D image and real image after statistical expression; and features of the (b) convolution and (c) deconvolution processes with the colour representing the origin of the deconvoluted values. For mapping palaeovalley patterns in an Australian desert landscape, as an example in this study, the input data uses the 2D MrVBF (an index calculated from globally available digital elevation model) with random white noise; the output is a 3D probability map of palaeovalley presence. For convenience of 3D convolution, the 2D input image (800×800×1)**
**is simply repeated in 10 layers to form a 3D input dataset (800×800×10); following a structure optimization by trial-and-error, the encoder is designed to contain 4 layers, with a width of 64, 32, 32 and 1 in each layer, respectively; the decoder contains 6 layers, with a width of 1, 16, 32, 32, 64, and 128, respectively; the discriminator contains 4 layers with a width of 128, 64, 32, 1, respectively.**

In the second step, the codes are converted into a 3D image of subsurface structure (a 3D array) by deconvolution (referred to

185 as 'decoder', Fig. 2a), which is a process involving a zero-padding before the convolution (Fig. 2c). The combination of

decoder and encoder forms a 'generator', linking input and output images. The generated 3D image is referred to as 'simulated image'.

To ensure that the simulated image is comparable to a real image, a voxel-wise independent heuristic criterion is minimized. The mean squared error ($L2$) between simulated and real images at all voxels is used as criterion to update the weight and bias in the decoder, which is expressed as:

$$L2 = \frac{1}{M} \|G(\mathbf{z}) - \mathbf{Y}\|^2, \tag{3}$$

where $M$ is the number of voxels in the simulated 3D image, $\mathbf{Y}$ is the real image, $\mathbf{z}$ is the code generated from the encoder, and $G(\cdot)$ represents the convolutional calculations in the decoder (in the same form as Eq. 1).

However, if only a limited number of real 3D images are available to train the network, the use of a voxel-wise independent criterion often leads to an overfitting problem. Goodfellow (2014) proposed a generative adversarial network structure, which adds a 'discriminator' to convert simulated and real images to a vector, respectively, by an identical convolution process (Fig. 2a). Adversarial criteria are proposed, typically expressed by binary cross entropy functions as:

$$L3 = -\frac{1}{V} \log[D(G(\mathbf{z}))], \tag{4}$$

and

$$L4 = -\frac{1}{V} \log[D(\mathbf{Y})] - \frac{1}{V} \log[1 - D(G(\mathbf{z}))], \tag{5}$$

where $V$ is the size of the output vector via the discriminator, and $D(\cdot)$ represents the calculations (Eq. 1) in the discriminator. The weights in the discriminator are trained to minimize $L4$, which attempts to distinguish the vectors generated from the real and simulated 3D images. The weights in the generator are trained to minimize $L3$, which attempts to fool the discriminator to be unable to identify the vector generated from the simulated 3D image. In such a way, the generator can produce images aligned with the real image in terms of probability structure (Goodfellow et al., 2014).

Finally, while the loss function $L4$ is minimized to optimize the weights in the discriminator, a comprehensive loss function combining $L1$, $L2$ and $L3$ is employed to optimize the weights in the generator, which is expressed as (Wu et al., 2016):

$$L_g = a \cdot L1 + b \cdot L2 + c \cdot L3, \tag{6}$$

where a, b, c are the coefficients on each loss function. This loss function makes it convenient to vary the neural network structure between semi-supervised learning with additional adversarial neural network by defining coefficient c as non-zero value, and supervised learning with merely autoencoder neural network with c as zero.

The hyperparameters (including the width, depth, filter size and the coefficients in generator loss functions, etc.) defining the neural network structure, are determined by trial-and-error tests (Supplementary materials). Weight and bias in generator and discriminator are trained to minimize $L_g$ and $L4$ using the stochastic gradient descent algorithm, referred to as adaptive moment estimation (ADAM) (Kingma and Ba, 2014). We implemented the above convolution neural network using the TensorFlow Python library (Abadi et al., 2016). Once the neural network is trained, the 'generator' in the network (Fig. 2a) is used independently to generate 3D subsurface structures from the 2D land surface observations.

## 3 Results

The effectiveness of our deep-learning model is tested on predicting 3D palaeovalley patterns in the Anangu Pitjantjatjara Yankunytjatjara (APY) lands, part of the Musgrave Province of South Australia (Fig. 3a and 3b). As discussed earlier, the palaeovalley networks in this region are remnants of the Late Mesozoic to Early Cenozoic inset valleys with coarse-to-fine grained sands infill, which is covered by a thin and variable Quaternary eolian sediments (Magee, 2009) (Fig. 3b). The 3D structure of a palaeovalley was interpreted from an airborne electromagnetic (AEM) survey (Soerensen et al., 2016). AEM data of sufficient spatial granularity (Line spacing of <400 m) to effectively define the spatial extent of near surface aquifer systems only exists in a limited number of prospective areas within close proximity to isolated townships. Our previous work evidenced that the palaeovalley geometry is correlated to the contemporary valley pattern in this region (Jiang et al., 2019) (compare Fig. 4a and 4b). The present-day valley pattern has been well defined by the Multiple-resolution Valley Bottom Flatness (MrVBF) index based on the slope calculated from the original 1-arc-second (around 30 m) SRTM-derived digital elevation model (details in Gallant and Dowling, 2003). Across the palaeovalley domain, MrVBF ranges between 4 (25[th] percentile) and 7 (75[th] percentile), with a median value of 6 (Fig. 4c).

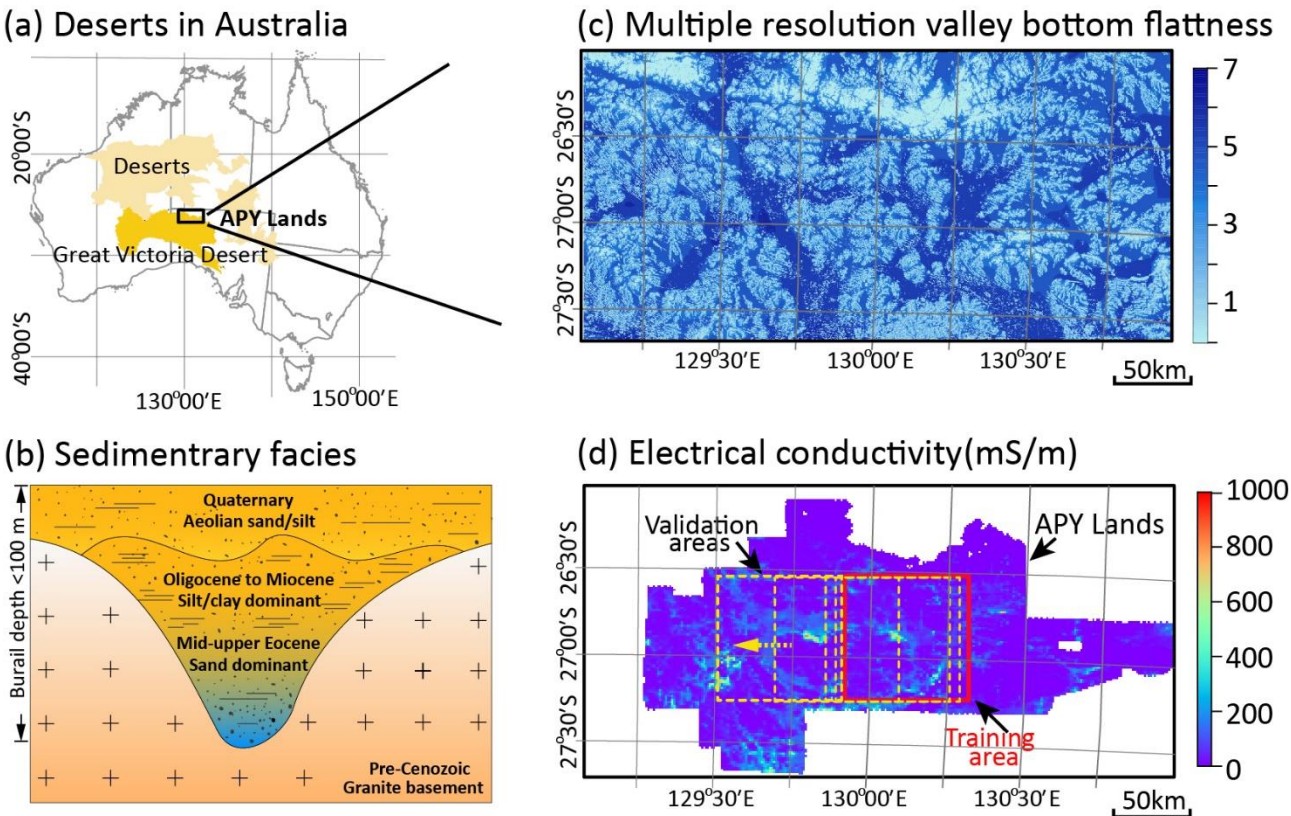

**Figure 3. Datasets for delineating 3D palaeovalley in the Anangu Pitjantjatjara Yankunytjatjara (APY) Lands of South Australia: (a) location of the largest deserts in Australia and (b) general conceptual model of palaeovalley sedimentary facies revealed by over 90% borehole logs, (c) multiple resolution valley bottom flatness index, and (d) electrical conductivity (at depths of 30 to 40 m with a horizontal resolution of 400 m) inferred by airborne electromagnetic surveys in the APY Lands, forming an indicator of palaeovalley presence.**

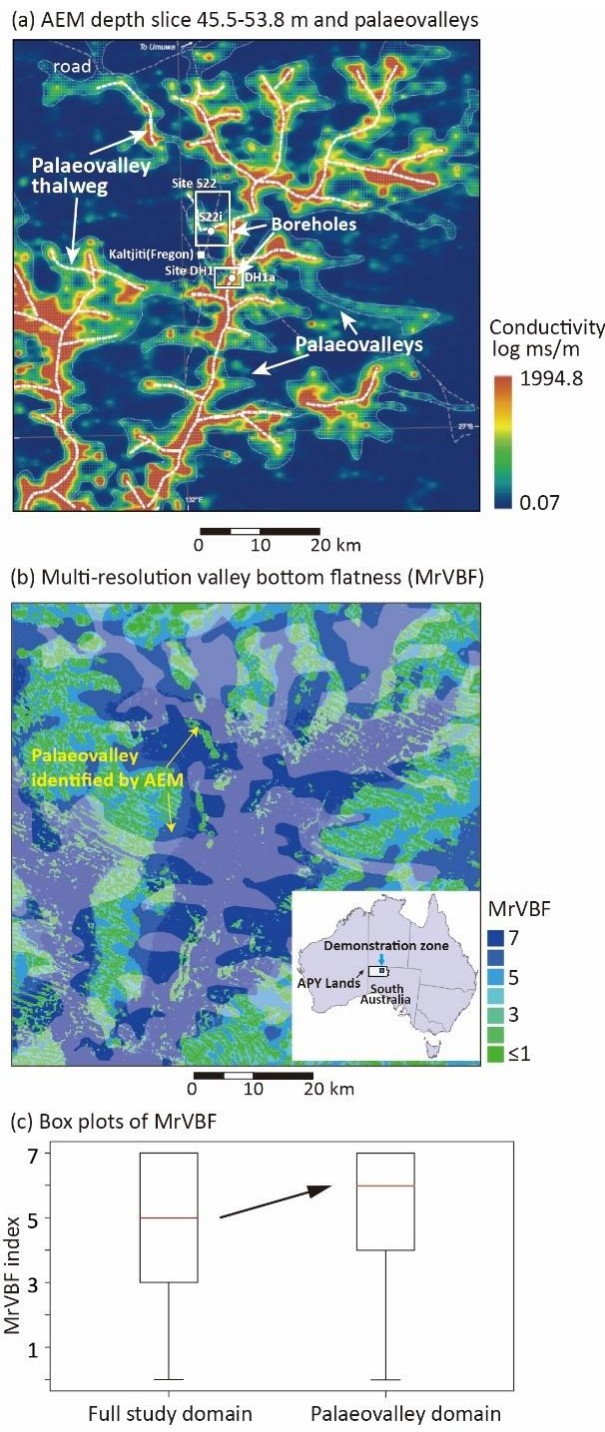

**Figure 4. (a) Air-borne electromagnetic (AEM) depth slice showing conductive palaeovalley areas, (b) corresponding MrVBF map with palaeovalley overlay, and (c) boxplot (interquartile range, median, and minimum value) of MrVBF across full study domain and for palaeovalley domain only.**

The MrVBF index exists across the entire Australia continent, whereas the AEM data coverage at a high resolution exists across a limited areas only, commonly confined to areas of mineral exploration interest or to areas prospective for groundwater (http://www.ga.gov.au/about/projects/resources/continental-geophysics/airborne-electromagnetics). The relationship between MrVBF and palaeovalley presence is complex, as palaeovalleys identified by AEM interpretation occurred in those zones with low MrVBF in addition to the high MrVBF that indicate the present-day valley (Fig. 4c). Our neural network model thus establishes a relationship between the MrVBF index and the AEM-derived 3D palaeovalley structure. This relationship is then used to predict the 3D palaeovalley structure in those areas with only MrVBF data but without the AEM dataset. For the method verification, both the training and prediction are conducted in the area where AEM data are available. Note that the weights in the neural network are determined based on the training area. The AEM data in the other areas are only used to test the predictive capability of the trained neural network.

The dataset employed for model verification includes the 1-second-arc resolution 2D MrVBF index across the entire model domain (Gallant and Dowling, 2003), and a 3D electrical conductivity dataset (400-m horizontal and 10-m vertical resolution) interpreted from a combined SkyTEM and TEMPEST time domain AEM survey across the APY Lands (Soerensen et al., 2016). These data are available from the CSIRO Data Access Portal (Munday, 2019). For the convenience of convolution operations, the MrVBF with 1-second-arc resolution available is normalized into the values ranging from zero to unity and is pre-smoothed into a spatial resolution of 100 m, by a 3×3 average filter. Considering that high bulk electrical conductivity values (EC) are a proxy for palaeovalley presence in contrast to the low EC of the underlying bedrock (Jiang et al., 2019; Munday et al., 2013; Taylor et al., 2015), the occurrence of palaeovalleys in this study is defined by what is termed as an AEM-derived Palaeovalley Aquifer Index (PAI):

$$\text{PAI} = \frac{\log_{10}(EC) - \log_{10}(EC)_{min}}{\log_{10}(EC)_{max} - \log_{10}(EC)_{min}}, \tag{7}$$

where *max* and *min* represent the maximum and minimum logarithm of EC values over the entire dataset, respectively. PAI ranges from 0.0 to 1.0 and is calculated in the first 100 m depth at the AEM-surveyed area, which is considered as a ground-truth 3D probability map of palaeovalley occurrences with a spatial resolution of 400 m×400 m×10 m. The effectiveness of the proposed model lies in predicting a 3D palaeovalley pattern equally well as that derived from AEM-derived EC values which represent the bulk geo-electric properties of the sediment infill, rather than specific lithofacies comparable to those interpreted from downhole logs.

A neural network simulator is established and trained to relate the AEM-derived PAI (output image) with 2D MrVBF data (input image). The training dataset covers part of the APY Lands (6,400 km[2]) (hereafter referred to as 'training area'). Both loss functions for discriminator and generator were monitored when training the model to verify the network being trained sufficiently (Supplementary materials). Training of the network under 10,000 iterations on a high-performance computer

(Tesla P-100-SXM2-M-16GB) required 100 to 150 minutes of computation time. Once trained, generating of 3D image from 2D MrVBF required less than five seconds on a desktop computer.

An area 80 km west of the training area is first used to validate the trained neural network in generating 3D PAI. In both the training and validation domains, the palaeovalley geometry in each layer is generally comparable to the surface valley geometry indicated by the MrVBF index at land surface (compare Fig. 5a with 5c and 5e, and Fig. 5b with 5d and 5f), with varying width at different depths. A horizontal transect or section through the middle of the study area for both training and validation areas illustrates the good correspondence between simulated and real PAI. Note the normalized MrVBF is typically at its maximum value everywhere the palaeovalley has a significant depth (Fig. 5g and 5h).

A comparison of the PAI error with the surface valley pattern in the validation domain (Fig. 6) shows that the spatial distribution of the largest prediction errors is rather random, with some concentration at the boundaries of modern-day valleys. This is because the convolution processes in the proposed model may smooth the conductive units adjacent to the resistive units, and the margins of conductive palaeovalleys gets distorted. The other possible source of error could be linked to the inversion procedure adopted with the AEM data, where a 1D code was employed. On the margins of the palaeovalleys, we may be encountering 2D or 3D effects that are poorly modelled with the 1D approach. The error distribution in the validation domain is independent from the modern-day valley geometry in the training area, suggesting that no overfitting problem occurs.

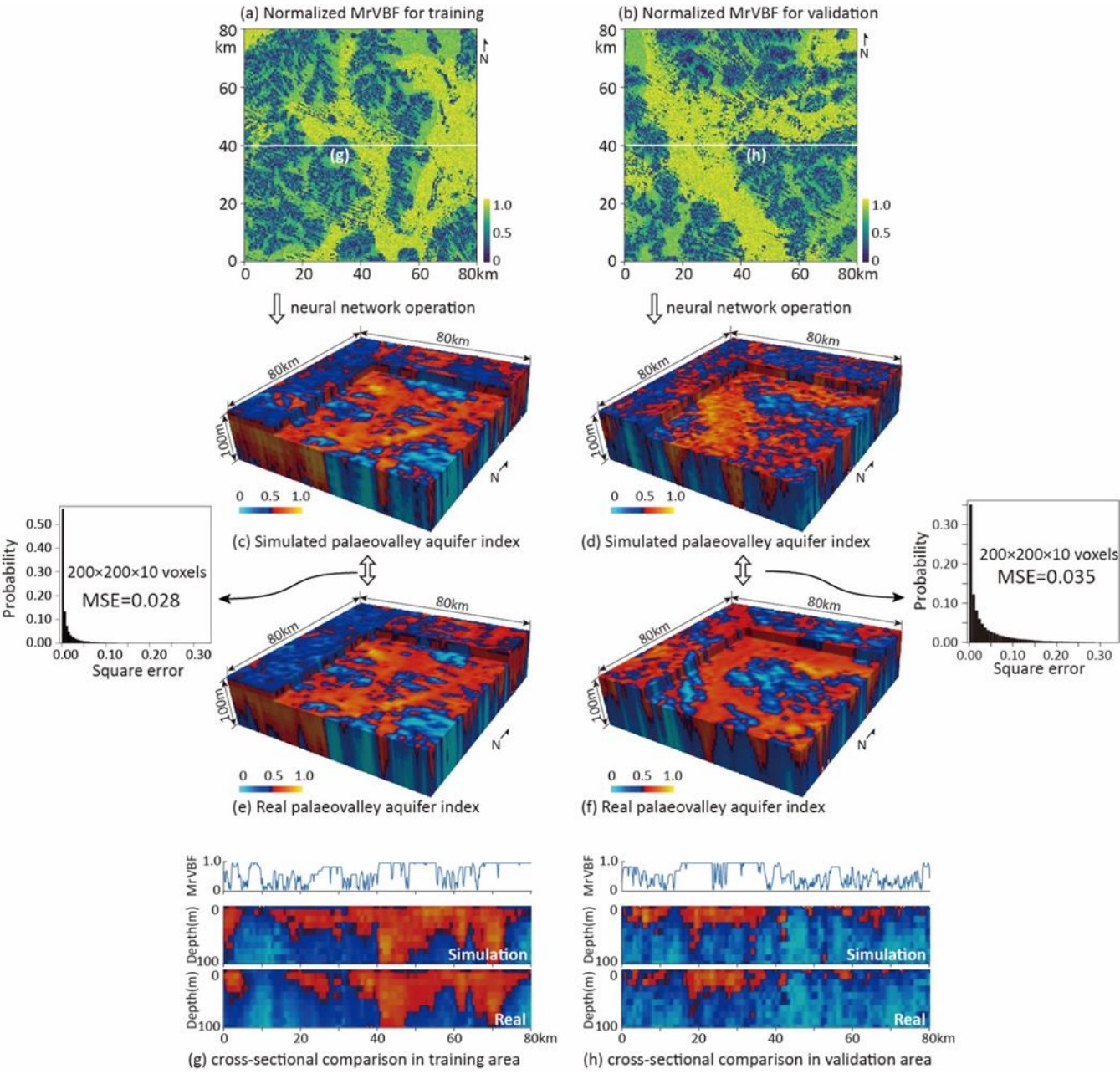

**Figure 5. Normalized multiple resolution valley bottom flatness (MrVBF) (a and b) converted to the 3D Palaeovalley Aquifer Index (PAI) in the training area (c) and validation area (80 km west to the train area) (d) by the neural network simulator, compared with AEM-derived PAI (ground truth data) (e) and (f) generated from airborne electromagnetic surveys. Transects of simulated and real PAI and corresponding normalized MrVBF for training area (g) and validation area (h).The trained neural network with the squared error < 0.10 across 99% of the training zone (a total of 200 ×200 ×10 voxels), results in a PAI error < 0.10 across 93% of this validation zone, with <1% of this validation zone having errors exceeding 0.20.**

The statistics of squared errors between the simulated 3D PAI and real PAI are calculated at all 200×200×10 voxels. As shown

in Fig. 5, the squared error in the training dataset is below 0.1 for 99% of the training domain and with a mean value of about

0.03, and the squared error of the predicted 3D PAI is well below <0.1 for 93% of the validation domain, with a mean squared

error of about 0.04. The patterns of the generated palaeovalley in both horizontal and vertical directions align with those

inferred from the AEM-derived PAI. This indicates that the deep-learning neural network structure developed here is capable

of incorporating the relationships between the MrVBF and the buried palaeovalley patterns, and allowing for high-quality

predictions beyond the training area.

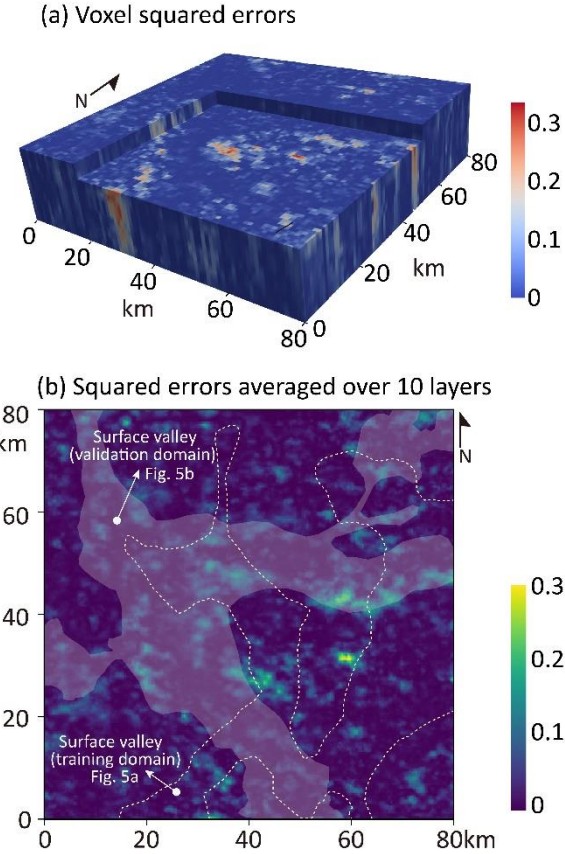

**Figure 6. (a) 3D distribution of squared errors between simulated PAI and real PAI in the validation domain, and (b) plan view of the mean of squared errors from ten layers, overlain by the surface (modern-day) valley (validation and training domains). The large errors, to some extent, focus on the edge of modern-day valley in the validation domain, but are unrelated to the modern-day valley in the training domain, suggesting that the overfitting does not occur.**

## 4. Discussion

### 4.1 Neural network with and without fully connected layer

The traditional convolution neural network is often ended by a fully connected layer in the encoder (e.g. Wu et al., 2016), where all the elements in previous layer are connected to every code in the output layer by matrixes multiplication. Such an

operation helps adequately fuse the input information for prediction. In this study, a 3D image with size of 25×25×5 is employed for the final output layer of the encoder (Fig. 2), without a fully connected layer. For comparison, a fully connected layer with a vector of 3125 (25×25×5) elements is employed as well. As shown in Fig. 7, both models can be trained to generate the palaeovalley in the training domain successfully (Fig. 7a to Fig. 7c and 7b, respectively). However, with a fully connected layer, the trained model fails to generate palaeovalleys in the validation domain. Under an alternative MrVBF as input (Fig.

7d), the predicted palaeovalley has a geometry very similar to that of the training domain (compare Fig. 7e and 7d). This suggests an apparent overfitting, caused by the fully connected operation fusing the input MrVBF globally.

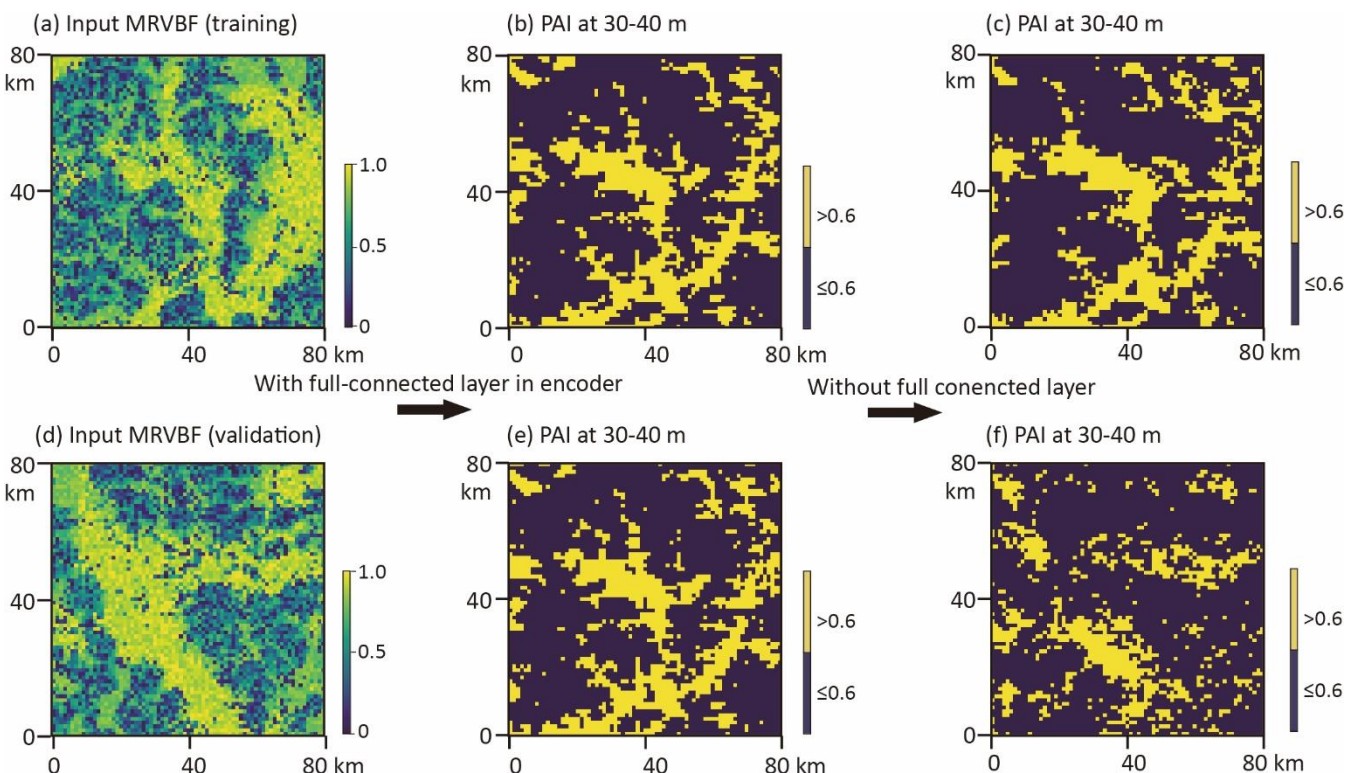

**Figure 7. Input MrVBF in (a) training area and (d) validation area, and (b) and (e) the generated PAI at depth of 30-40 m with fully connected operation in the encoder, and (c) and (f) without fully connected operation.**

Alternatively, the model without a fully connected layer can predict well the palaeovalley following the MrVBF pattern. Without the fully connected layer, the convolution processes with 3D filter addressed the local relationship of MrVBF and PAI. The correlation scale is determined by the size of the filter; the lager the filter, the larger the correlation scale addressed. The filter size can be determined by a trial-and-error test, according to the misfit between the predicted geological variable and

the ground truth data in both training and validation domains. In this study, a filter with a size of 4×4×2 is employed for the encoder and discriminator, while a filter with size of 5×5×2 is employed for the decoder (details in the supplementary materials).

Training and validation suggest that using relatively small filter and removing the fully connected layer to under-parameter the neural network model helped reducing the overfitting risk. Although the performance of the neural network model with this given structure is acceptable, relatively large errors still occur at the boundaries of the palaeovalley where the MrVBF values vary sharply. This is because local convolution potentially broadens the influence of large MrVBFs; adaptive optimization of filter size in each convolution layer potentially solves this problem.

### 4.2 Adversarial neural network versus autoencoder neural network

Furthermore, another 19 validation areas west to the training domain (Fig. 3d) are used to monitor the decay in the accuracy of predicted palaeovalley patterns. This is done using two different models: semi-supervised learning with additional adversarial neural network and supervised learning with only autoencoder neural network (controlled by coefficient c in Eq. 6).

As shown in Fig. 8, an extremely small error (<0.01) can be achieved when constructing palaeovalleys in the training area by supervised learning using only the autoencoder neural network. The mean error resulting from the semi-supervised learning with additional adversarial network is higher, i.e. about 0.03. Within the 19 validation areas, the mean squared errors in predicting palaeovalley patterns by both neural networks are well below 0.04.

While the autoencoder learning generally performs better than the adversarial learning in terms of mean squared errors, its prediction errors (especially the 95% quantile) increase much faster with the separation distance between validation and training areas. This indicates that using the autoencoder only can potentially lead to very large errors (or poor predictions) in case of discrepancies between training and prediction areas. We hypothesize that these errors are due to model overfitting in the case of using only the autoencoder learning. To confirm this, we conduct an overfitting test based on a synthetic ground being a random MrVBF input following a uniform distribution (i.e. non-informative) ranging from 0 to 1, which should result in a uniform PAI distribution. As shown in Fig. 8c, a uniform PAI can be generated by adversarial learning, while the predicted PAI by using only the autoencoder learning results in structured patterns. This means that the weights trained by purely supervised learning inherit too much information hidden in the training dataset, which is inflexible in predicting 3D palaeovalley patterns with strong variations from the input image. On the other hand, adversarial learning is much more robust to discrepancies and the accuracy decays only slightly in predicting 3D structures in areas further away from the training area, which is a highly desired property in real world applications.

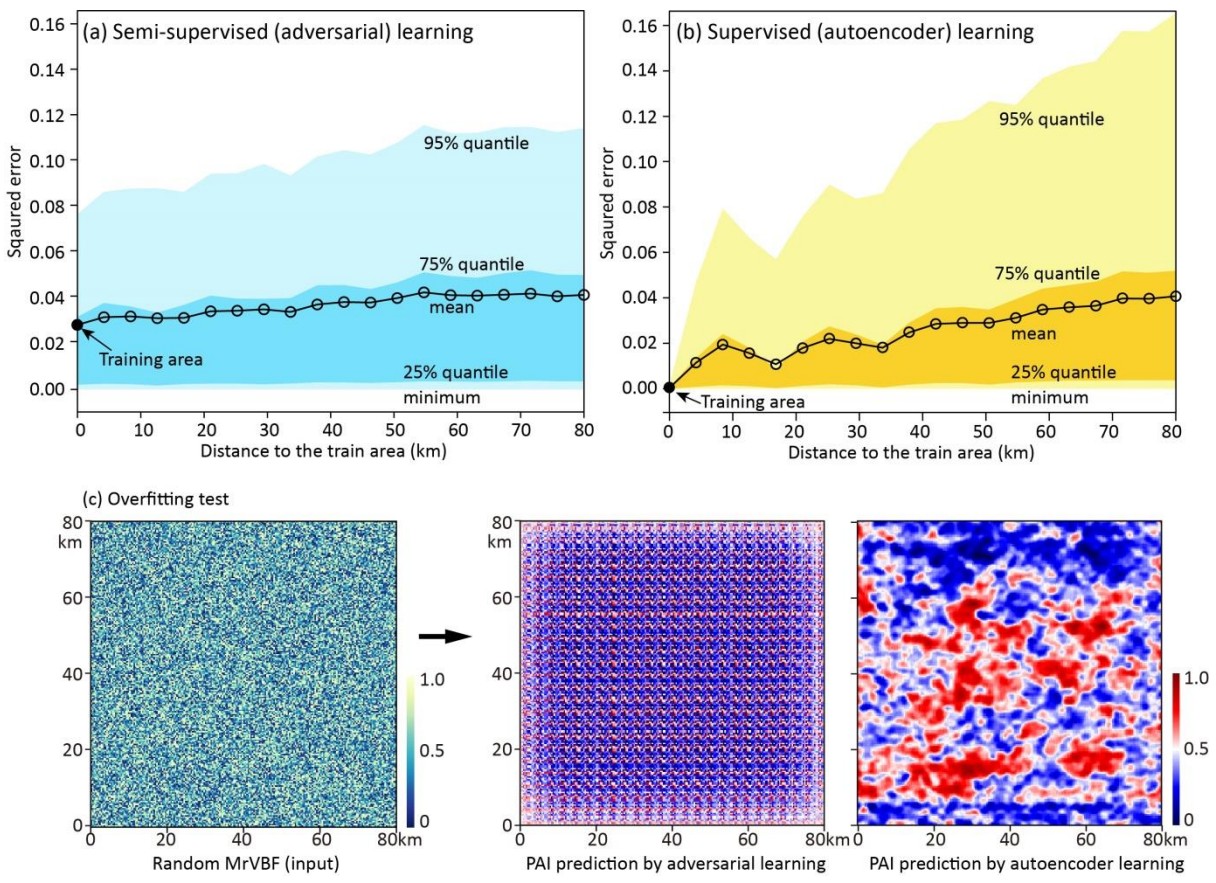

**Figure 8. Squared errors between the true 3D palaeovalley aquifer index (PAI) directly calculated from AEM-derived electrical conductivity, and PAI predicted by (a) autoencoder neural network using supervised learning and (b) adversarial neural network using semi-supervised learning in the areas west to the training area, with separation distance varying from 0 to 80 km; and (c) an overfitting test with a random 2D MrVBF as input to predict PAI at depth of 30-40 m following adversarial learning and only autoencoder.**

### 4.3 Generalisation

The geomorphological evolution of palaeovalley systems is never straightforward. Our study site, for instance, remains tectonically active, although not in a manner where more recent tectonism leads to hills and ranges. Rather the neotectonism leads to changes in the hydraulic conductivity of aquifer systems. In some instances this results in marked changes in the conductivity structure across faults which transect palaeovalley systems (see, for example, Munday, 2020a; Munday et al., 2001). Furthermore, application of AEM for mapping buried valley systems has been successful in several other areas across Australia, each with their own evolutionary intricacies (Davis et al., 2016; Magee, 2009; Roach et al., 2014) with the key commonality being a very low topographic gradient. AEM has also been used in northern Europe, Canada and the US for similar purposes, albeit with different electrical conductivity structures. The application of AEM to map palaeovalley systems in many parts of the world has been successfully demonstrated.

Indeed, palaeovalleys occur beneath the glaciated landscapes of Northern Europe, Canada, and the Northern USA. When filled with coarse-grained permeable sediments, these valleys – as their Australian counterparts - represent potential sources of

365 groundwater. In Northern Germany, shallow strata deposited during Quaternary times developed into palaeovalley systems characterized by a ground floor topography filled by fine grained marine and glacio-marine sediments. In these systems, AEM was successfully used to derive a detailed 3D geological model of the 350-m deep and 0.8-2 km wide valley infill (Siemon et al., 2006). Similar buried valleys with heterogeneous infill have been reported for Denmark, with typical dimensions of 0.5-4 km wide, and 25-350 m deep; their lengths varies from roughly 30 km for onshore structures to 100 km for offshore systems

(Høyer et al., 2015; Jørgensen et al., 2003). In our study site, the burial depth of palaeovalley infills ranges between 5.0 to 250 m, with the typical width from 0.1 to 2 km.

In southern Manitoba, Canada, Oldenborger et al. (2013) used a combination of airborne time-domain electromagnetics, electrical resistivity and seismic reflection to map the complex buried valley morphology with nested scales of valleys at a level of detail sufficient for groundwater prospecting, modelling and management. Korus et al. (2017) demonstrated that AEM

can be used effectively in environments like the glaciated Central Lowlands of Nebraska (USA) to identify sedimentary architectural units with a high degree of lithological heterogeneity. These systems were tens of meters deep and 100 m to more than 1000 m wide.

All these valley-infill systems are characterized by a multi-phase history of glaciation and buried valley genesis. The palaeovalley systems in our study area and the broader Musgrave Province/Great Victorian Desert also have a multi-phase

history, albeit with somewhat different processes across more extended timescales. Importantly, geophysical inversions across a wide range of palaeovalley systems have consistently delivered realistic geologic profiles, albeit defined by the geo-electrical properties of the fill materials and the water contained therein (Davis et al., 2016; Magee, 2009; Roach et al., 2014; Soerensen et al., 2016). They thus form a sound basis for subsequent developments such as deep-learning model for predictive purposes. Among the many potentially suitable geoscientific data sets for deep-learning-based prediction of palaeovalley boundaries and

internal structure, topographic information was shown in this study to be a suitable predictor across a large test area (6400 km$^2$).

Valleys are, by definition, low points in the landscape and therefore topographic information is pivotal when mapping palaeodrainage patterns. In Australia, with its long-term tectonic stability, the topography of drainage systems has survived for very long periods of time. The presence of Mesozoic- Cenozoic pre-existing valleys has survived in the new landscape,

because both erosion and deposition rates are extremely slow. These factors have combined to preserve many ancient Tertiary palaeodrainage patterns and in most instances palaeovalleys are still actual valleys, eventhough relief is subdued. Digital elevation models are very effective in recognising such Tertiary palaeovalleys and related features because the modern and Tertiary geomorphologies are usually related, both spatially and genetically (Magee, 2009).

Further characteristics of the palaeovalley landscape of the Musgrave Province are the extensive aeolian sandplains and sand dunes that overlay the valleys; groundwater calcrete and gypsum-rich playa sediments are evident in palaeovalleys where sand dunes are absent (Magee, 2009). These sand dunes were deposited around 200, 000 years ago (Krapf et al., 2019). The thickness of these sand deposits varies, but drillhole investigations revealed the boundary between overlying sandplains and palaeovalley to be around 30 m for major valleys to 10 m for tributary channels (Krapf et al., 2019). As a result, the palaeovalleys have only a subtle surface expression in today's landscape. As we have demonstrated, detailed topographic data such as high-resolution MrVBF can be successfully used to detect such subdued surface expressions and infer the presence of buried systems.

In summary, palaeovalley relief is minimized and concealed by infill material, overlying sediments and the formation of playas (salt lakes). As a result, DEMs and its derivatives like MrVBF do not always permit the direct interpretation of palaeovalley boundaries, while the palaeochannel facies are even more difficult to infer (Hou et al., 2000). However, palaeodrainage systems in our study area mostly coincide with topographic lows characterized by MrVBF values between than 4 and 7 (inter-quartile range) (Fig. 4c).

So far, our deep-learning model has been tested and validated in the Great Victorian Desert only, noting the areas for training and validation were considerable in size, each 6400 km$^2$. Based on the characteristics of the palaeavalleys and the topographic features of the surrounding terrain discussed above, potentially suitable areas for further model testing can be identified. Note that the proposed model is not restricted to topographic input parameters only; any parameter that can be correlated with palaeovalley structure and features has potential to be used for predictive purposes. Therefore, the model developed here mainly serves as a generic framework that has applicability also in other areas, with input data not restricted to topographic information but also including remote sensing and geophysical data (Hou and Mauger, 2005).

## 5. Conclusions

This study developed an efficient and reliable adversarial convolutional neural network model for generating 3D subsurface structures directly from 2D land-surface data. The proposed generic structure of the convolutional neural network was composed of an 'encoder' to fuse 2D input data into low-dimension codes following a normal distribution, a 'decoder' to nonlinearly map the low-dimension codes into 3D subsurface images, and a 'discriminator' to statistically express the generated and real subsurface image into a vector for adversarial semi-supervised learning based on a single training image.

The neural network was tested for mapping the 3D structure of buried palaeovalley systems in the northeast Great Victoria Desert, Australia. Training and validation involved using the multiple resolution valley bottom flatness (MrVBF) (input) and 3D palaeovalley aquifer index (PAI) (output) on an area of 80×80 km$^2$. The neural network trained with errors <0.1 across 99% of the training domain can predict 3D PAI with errors <0.1 at over 90% of the validation zones.

The performance of the deep-learning neural network for 3D subsurface structure imaging has applications as a generic novel tool for making better use of existing multiple-support 2D land surface observations (e.g. surface geology map, digital elevation data, and remote sensing) for better management of limited resources such as groundwater.

**Codes/Data Availability**

The original data of MrVBF and AEM were provided by John Gallant and CSIRO, respectively, and are available freely from CSIRO Data Access Portal https://doi.org/10.4225/08/5701C885AB4FE and https://doi.org/10.25919/5d0868d48591e. The codes for neural network developed in Tensorflow are now provided in https://doi.org/10.7910/DVN/DDEIUV.

**Acknowledgement**

Funding support for this study was provided by the National Key R&D Program of China (2018YFB1501803), and the International Postdoctoral Exchange Fellowship Program (2017) from China Postdoctoral Council in combination with CSIRO funding through the Land and Water Business Unit and the Future Science Platform Deep Earth Imaging. We thank Guillaume Rongier and Hoel Seille for the internal review and constructive suggestions on this article.

**Author Contribution**

ZJ developed the model code and performed the modelling; DM, LG, TM, GM and LP collected the data and evidence, and provided constructive feedbacks for the improvement of the model; ZJ and DM prepared the manuscript with contributions from all co-authors.

**Competing interests**

The authors declare that they have no conflict of interest.

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
