# Peer review of "Sub3DNet1.0: A deep learning model for regional-scale 3D subsurface structure mapping"

_Geoscientific Model Development, 2020_

## Referee Comment (RC1) · Anonymous Referee #1 · 23 Nov 2020

General comments: This manuscript proposes a deep CNN with joint autoencoder and adversarial structures to predict the probability of subsurface palaeovalleys (derived from airborne electromagnetic data) using 2D land surface tomography. It has been claimed that the trained model "produces a square error < 0.10 across 93% of the validation areas". The prediction error contradicts the conclusion of a reliable model in reconstructing 3D palaeovalley patterns. This is consistent with the results in Figure 3. If we compare Figures 3d and 3f, it is quite clear that many structures that are present in the real 3D image are missing from the simulated image and indeed these two images are not similar. On the other hand, Figure 3c and 3e are very similar (Training set). This simple visual comparison reveals that the trained model is very overfitted and contradicts the claim of similar performance in training and validation areas (Abstract:

[Figure]

"The trained neural network has a maximum square error < 0.10 and produces a square error < 0.10 across 93% of the validation areas"). I highly recommend the authors to provide more evidence on the performance of the proposed model in validation areas. A 3D map showing the spatial distribution of errors (for both validation and training areas) would be useful.

Specific comments: The proposed model is used for subsurface structure mapping. Sub3DNet might be a better name for the model. It is good to discuss some of the limitations of the deep CNN models. For instance, too many structures are available (e.g., number of convolutional and pooling layers) and it is not clear which structure is the best for the study presented in this manuscript.

Technical corrections: No technical correction is needed at this stage.

---

## Author Comment (AC1) · 28 Nov 2020

We highly appreciate your time in reviewing our manuscript. The relevant comments have improved the quality of the manuscript. We now revised the manuscript accordingly. The detail responses are listed below. The modifications in the article are marked in the annotated manuscript (in the supplement files).

Q1. This manuscript proposes a deep CNN with joint autoencoder and adversarial structures to predict the probability of subsurface palaeovalleys (derived from airborne electromagnetic data) using 2D land surface tomography. It has been claimed that the trained model "produces a square error < 0.10 across 93% of the validation areas". The prediction error contradicts the conclusion of a reliable model in reconstructing 3D

palaeovalley patterns. This is consistent with the results in Figure 3. If we compare Figures 3d and 3f, it is quite clear that many structures that are present in the real 3D image are missing from the simulated image and indeed these two images are not similar. On the other hand, Figure 3c and 3e are very similar (Training set). This simple visual comparison reveals that the trained model is very overfitted and contradicts the claim of similar performance in training and validation areas (Abstract: "The trained neural network has a maximum square error < 0.10 and produces a square error < 0.10 across 93% of the validation areas"). I highly recommend the authors to provide more evidence on the performance of the proposed model in validation areas. A 3D map showing the spatial distribution of errors (for both validation and training areas) would be useful.

Re: Partially agree and changes made.

(1) Consistence. We now carefully checked through the calculations and the resulting values, and make sure that the expression of "produces a square error < 0.10 across 93% of the validation areas" is consistent with the values used to draw Fig. 3. In the training domain, it was calculated that the 0.991 quantile of errors is equal to 0.1. The expression "The trained neural network has a maximum square error < 0.10" is now reformulated as "The trained neural network has a square error < 0.10 across 99% of training domain" in both the abstract and the context (e.g. Line 13, 134 and 145), to better express the findings.

(2) Overfitting. We now draw the 3D distribution of errors in the validation domain and also a plan view of errors averaged over ten layers (now Fig. 4). The error distribution is compared to the modern-day valley pattern suggested by the MrVBF in both validation and training domains, because the paleovalley geometry inherits the pattern of modern-day valley (comparing Fig. 3a and 3c, 3b and 3d). It is illustrated that the distribution of large errors in the validation domain is unrelated to the modern-day valley geometry in the training domain, but some concentrate on the boundaries of surface valley in the validation domain. The former confirmed that no overfitting problem occurs. The latter is induced by the convolution processes itself. This is now expressed in Line 147-155.

A futher comparion between the neural network without and with fully connected layers is now added in Line 158-179. As a result, using relatively small filter and removing the fully connected layer to under-parameter the neural network model helped reducing the overfitting risk.

Q2. The proposed model is used for subsurface structure mapping. Sub3DNet might be a better name for the model.

Re: Agree. We now changed the model name as Sub3DNet.

Q3. It is good to discuss some of the limitations of the deep CNN models. For instance, too many structures are available. (e.g., number of convolutional and pooling layers) and it is not clear which structure is the best for the study presented in this manuscript.

Re: Agree. We now added a Discussion section in Line 158-179, to (1) compare the proposed CNN with the traditional structure, (2) clarify the limitation of CNN model. We also put the details of the CNN structure optimization (e.g. depth, width, filter sizes of neural networks) in the support materials.

Please also note the supplement to this comment:
https://gmd.copernicus.org/preprints/gmd-2020-106/gmd-2020-106-AC1-supplement.zip

---

## Referee Comment (RC2) · Anonymous Referee #2 · 8 Dec 2020

Dear authors: Please do not be too discouraged by this less than positive review, since it may be due to some major misunderstandings that I outline below. I hope they are indeed misunderstandings and you can revise to resubmit, as I suspect your DL algorithm will ultimately be worthy of publication.

My review of this paper is not favorable mainly because, despite repeated readings of it, I am unable to identify the specific research problem that the authors are seeking to solve, and because the case study used to demonstrate their method appears to be trivial in the context of subsurface characterization. Although it is possible my reactions stem from major misunderstanding of the descriptions of the objectives, methods and results, I have spent decades of my career mapping and modeling paleochannels, including application of AEM and other geophysical methods, yet I am unable to reconcile the separation between what the authors are writing and what I would consider to be understandable or obvious contributions.

My trouble with the objectives and problem definition can be best illustrated by first considering the geologic system the authors seek to better map in 3D. 'Paleochannels' can take on a number of different meanings depending on the geologic setting, but from what I can decipher from the introduction, methods and Fig. 2, by 'paleochannels' the authors are referring to incised valley fill deposits like those depicted in Fig. 2b, where the channels are bounded not by adjacent fluvial facies, but by granite. Setting aside for the moment that this looks more like a basin and range style of geologic structure than a paleochannel, based on the vague descriptions in the paper, I can only construe that the flattest portions of the DEM shown in Fig. 2c represent the Quaternary alluvial bottomlands representative of the top of the apparent paleochannels (i.e., top of sed facies in Fig. 2b). If that is true, the reader's reaction is inevitably: "Why is this even considered a challenging problem? From the topography it is already obvious where these so-called channel deposits locate."

Summarizing the case study, it appears that the DEM already nicely identifies locations of the paleochannels, which apparently have been further characterized using AEM, presumably to better identify their depths or depth to bedrock perhaps. This raises the question of what is the problem the authors are attempting to address? If the problem is to better identify x-y locations of the so-called paleochannels, that would appear moot because the DEM already shows them, which also raises the question of why you need DL. If the problem is to better identify paleochannel or incised valley-fill depths, that has apparently already been done with AEM; and furthermore, if the purpose is to use the DL algorithm to map the paleochannels depths so that AEM would not be needed, that also does not appear to make sense because the authors have not established a relationship between the DEM flatness metric and paleochannel depths.

It is possible that if the authors can be more specific about the geology of these 'paleochannel' features that they are trying to map and about what specifically they are

trying to accomplish through the application of their ML methods, the above problems would be cleared up. As written, however, the manuscript lacks sufficient definition of the problem, description of their objectives, and description of how their research satisfies those objectives.

I am attaching an annotated PDF file showing some edits and comments in the manuscript. This is best viewed as two-facing pages, where the left page shows the manuscript page with index numbers, and the right page shows the indexed comments and edits.

Please also note the supplement to this comment:
https://gmd.copernicus.org/preprints/gmd-2020-106/gmd-2020-106-RC2-supplement.pdf

––––––––––––––––––––––––––––

**Supplement:**

**A deep learning model for regional-scale 3D subsurface structure mapping**

Zhenjiao Jiang [1,2], Dirk Mallants [2], Lei Gao [2], Gregoire Mariethoz [3], Luk Peeters [4]

1 Key Laboratory of Groundwater Resources and Environment, Ministry of Education, College of Environment and Resources, Jilin University, Changchun, 130021, China

2 CSIRO Land & Water, Locked Bag 2, Glen Osmond, SA 5064, Australia

3 University of Lausanne, Faculty of Geosciences and Environment, Institute of Earth Surface Dynamics, Lausanne, Switzerland

4 CSIRO Mineral Resources, Locked Bag 2, Glen Osmond, SA 5064, Australia

*Corresponding author:* Zhenjiao Jiang (jiangzhenjiao@hotmail.com)

**Abstract.** This study introduces an efficient deep learning approach based on convolutional neural networks with joint autoencoder and adversarial structures for 3D subsurface mapping from surface observations. The method was applied to delineate palaeovalleys in an Australian desert landscape. The neural network was trained on a 6,400 km$^2$ domain by using a land surface tomography as 2D input and an airborne electromagnetic (AEM)-derived probability map of palaeovalley presence as 3D output. The trained neural network has a maximum square error < 0.10 and produces a square error < 0.10 across 93% of the validation areas, demonstrating that it is reliable in reconstructing 3D palaeovalley patterns beyond the training area. Due to its generic structure, the neural network structure designed in this study and the training algorithm have broad application potential to construct 3D geological features (ore bodies, aquifer) from 2D land surface observations.

**1 Introduction**

Imaging the Earth's subsurface is crucial for the exploration and management of mineral, energy and groundwater resources; its reliability depends on the availability and quality of geological data. Although the amount and quality of geological data obtained from borehole logs, geophysical prospecting and remote sensing has increased  over the past decades, their spatial distribution is highly uneven. Most data exist  in a limited number of highly-developed areas such as mining and oil fields.

Commonly used methods for modelling complex geological structures include geostatistical approaches such as sequential Gaussian or indicator simulation (Lee et al., 2007), transition probability simulation (Felletti et al., 2006; Weissmann and Fogg, 1999), or multiple-point simulation (MPS) methods (Hu and Chugunova, 2008; Mariethoz and Caers, 2014; Strebelle, 2002). However, they often present drawbacks such as  being inefficient in capturing essential features and patterns from very large training datasets, or presenting a high computational cost. A fast and reliable tool for high-resolution 3D subsurface imaging based on multiple-support big dataset  Deep learning approaches specialised in big data mining have the potential to fill this gap (Gu et al., 2017; Hinton and Salakhutdinov, 2006; Marcais and de Dreuzy, 2017). Applications in the geosciences include earthquake detection based on seismic monitoring (Mousavi and Beroza, 2019; Perol et al., 2018), or disaster recognition from remote sensing data (Amit et al., 2016; Längkvist et al., 2016), among others. A recent breakthrough in deep learning is the 2D to 3D image processing (Niu et al., 2018; Sinha et al., 2017; Wu et al.,

**Summary of Comments on gmd-2020-106-manuscript-version1_ed.pdf**

**Page: 1**

[Figure]

Number: 1 Author: Subject: Cross Out Date: 12/6/20 8:00:43 PM

Author: Subject: Sticky Note Date: 12/6/20 8:00:57 PM
??

Number: 2 Author: Subject: Cross Out Date: 12/6/20 7:59:57 PM

Number: 3 Author: Subject: Cross Out Date: 12/6/20 8:02:43 PM

Author: Subject: Sticky Note Date: 12/6/20 8:03:25 PM
"data poor" contradicts "rich/big".

Number: 4 Author: Subject: Cross Out Date: 12/6/20 8:01:53 PM

Number: 5 Author: Subject: Cross Out Date: 12/6/20 8:03:32 PM

Number: 6 Author: Subject: nserted Text Date: 12/6/20 8:01:49 PM
Improved

Number: 7 Author: Subject: nserted Text Date: 12/6/20 8:02:22 PM
analyzing

Number: 8 Author: Subject: Cross Out Date: 12/6/20 8:05:46 PM

Author: Subject: Sticky Note Date: 12/6/20 8:07:06 PM
This is most certainly not true of those methods, although one might need to use them more expertly (e.g., through zoning of the model region) when non-stationarities are present.

Number: 9 Author: Subject: nserted Text Date: 12/6/20 8:10:30 PM
would be beneficial (To authors: this is weak as a motivational statement, and distorts the truth about capabilities of existing methods to justify your method. It is sufficient for your method to produce improvements on the other methods, and you can rest your case on that premise rather than asserting the other methods not nearly as functional.)

Number: 10 Author: Subject: Highlight Date: 12/6/20 8:11:50 PM
You have not identified a gap, but rather a potential way of improving upon other methods. Rewrite to better describe the "gap" and what your method potentially does.

[revised manuscript text omitted]

Number: 1 Author:     Subject: Highlight     Date: 12/6/20  9:06:24 PM

This would appear to represent existing fluvial drainage characteristics. For this to be useful for 'training' the DL model there would have to be a mechanistic connection (geologic) between these surface features and the subsurface distribution of paleochannels. There is a big problem with this approach: modern geomorphic surface characteristics seldom represent or correlate to the morphology and distributions of subsurface facies or rock types.

Number: 2 Author:     Subject: Highlight     Date: 12/6/20  9:10:20 PM

Not clear. Do the valley bottoms in c correspond each to the type of channel and facies depicted in b? If yes, does that mean these are all incised into granite? In that case, the predictive geologic problem would appear to be trivial.

Number: 3 Author:     Subject: Highlight     Date: 12/6/20  9:08:11 PM

Is the point here to use the AEM results as a ground truth and demonstrate that you could do as good, or almost as good, without the AEM and just using your DL approach based on surficial information? Not clear.

which is considered as a ground-truth 3D probability map of palaeovalley occurrences with a spatial resolution
130  of 400 m×400 m×10 m.

[Figure]

Figure 2. Datasets for delineating 3D palaeovalley in the Anangu Pitjantjatjara Yankunytjatjara (APY) Lands of
South Australia: (a) location of the largest deserts in Australia and (b) general conceptual model of palaeovalley
sedimentary facies revealed by over 90% borehole logs, (c) multiple resolution valley bottom flatness index, and (d)
135  electrical conductivity (at depths of 30 to 40 m with a horizontal resolution of 400 m) inferred by airborne
electromagnetic surveys in the APY Lands, forming an indicator of palaeovalley occurrence.

A neural network simulator is established and trained to relate the [1]EM-derived PAI (output image) with 2D
MrVBF data (input image). The training dataset covers part of the APY Lands (6,400 km$^2$) (hereafter referred to
as 'training area'). Both loss functions for discriminator and generator were monitored when training the model
140  to verify the network being trained sufficiently (Supplementary materials). Training of the network under
10,000 iterations on a high-performance computer (Tesla P-100-SXM2-M-16GB) required 100 to 150 minutes
of computation time. Once trained, generating of 3D image from 2D MrVBF required less than five seconds on
a desktop computer.

An area 80 km west of the training area is first used to validate the trained neural network in generating 3D PAI.
145  The statistics of square errors between the simulated 3D PAI and real PAI are calculated at all 200×200×10
voxels. As shown in Fig. 3, the squared error in the training dataset is below 0.1 and with a mean value of about
0.03, and the squared error of the predicted 3D PAI is well below <0.1 for 93% of the validation domain, with a
mean squared error of about 0.04. The patterns of the generated palaeovalley in both horizontal and [2]vertical
directions align with those inferred from the AEM-derived PAI. This indicates that the deep-learning neural
150  network structure developed in this work is capable of incorporating the relationships between the MrVBF and
the buried palaeovalley patterns, and allowing for reliable predictions beyond the training area.

Number: 1 Author:     Subject: Highlight     Date: 12/6/20  9:11:10 PM

But you have not established why these should possibly be related.

Number: 2 Author:     Subject: Highlight     Date: 12/6/20  9:19:15 PM

The vertical extents appear to be extraordinarily thick, essentially extending over the total 100 m thickness of the modeled region. This does not seem realistic unless these paleo vallies are just incised valley fill deposits produced by long term hard-rock incision, followed by relatively simple aggradation within those vallies. As such, this does not appear to be a difficult predictive test for the method, and the problem you are attempting to solve (or demonstrate) is even less clear.

---

## Author Comment (AC2) · 14 Dec 2020

We highly appreciate your time in reading through this manuscript many times, and give us many constructive comments on the background presentation. We have revised the introduction and study area sections accordingly, to better articulate the research problems we aim to solve, among others. The detailed responses are listed below and the modifications are marked in the annotated manuscript (in the supplement file).

Q1. My review of this paper is not favorable mainly because, despite repeated readings of it, I am unable to identify the specific research problem that the authors are seeking to solve, and because the case study used to demonstrate their method appears to be trivial in the context of subsurface characterization. Although it is possible my reac-

tions stem from major misunderstanding of the descriptions of the objectives, methods and results, I have spent decades of my career mapping and modeling paleochannels, including application of AEM and other geophysical methods, yet I am unable to reconcile the separation between what the authors are writing and what I would consider to be understandable or obvious contributions.

Reply:We agree that the research problem was not well articulated and have made changes to support our claim the work is novel with many practical applications to better mapping of shallow subsurface features and their geometries.

We rewrote the Introduction to better articulate the novel contribution in the method development. The major changes included are:

(1) in the first paragraph (Lines 22-31), we now describe that big data sets on geology and geomporphology are globally available either as land surface observations (typically remote sensing and topographical data and their derivatives), or regionally available in a limited number of highly-developed mining and oil fields (e.g., downhole, surface and airborne geophysical interpretations). In Australia, the former are readily available at low cost, while the latter are often non-existing and expensive in remote desert areas where groundwater for town supply relies on access to shallow aquifers (Munday et al., 2020a). In their study, Munday et al. (2020) interpreted 17,000 line km of airborne electromagnetic (AEM) data covering an area of about 30,000 km2, a fraction of the 422,000 km2 Great Victoria Desert in central Australia. With a AEM line spacing of 2 km, with smaller infill areas where line spacing was reduced to 250 and 500 m to provide greater detail of the subsurface electrical conductivity, accurate mapping of palaeovaleys was achieved (Munday et al., 2020b). Application of such high-resolution data to much larger areas like the Victorian Desert would be cost prohibitive. Our goal is therefore to develop an efficient and generic tool to express the relationship between an easy-to-obtain dataset and a more costly dataset for the specific purpose of detecting palaeovalley features that would facilitate the discovery of new groundwater resources in arid and semi-arid regions. In other words, we seek to develop a novel

method that uses AEM only for model development on a small training area while the application (i.e. detection of palaeovalleys across large areas) uses readily available landsurface information that otherwise (i.e., without AEM coupling through a training procedure) would have had little value for palaeovalley detection.

(2) in the second paragraph (Lines 38-53), we describe the limitation of the existing methods. For example, the traditional geostatistical methods are skillful in interpolation but not in extrapolation. MPS is powerful in delineating complex subsurface structures, but its effect depends on the availability of the training data. These methods are developed and employed based on the single-support dataset, that is, the data types employed to define spatial relationship is presumed to be the same as those data types employed to predict the subsurface geo-body. They are often inefficient in capturing essential features and patterns from large and multiple-support datasets, or can do so only at a high computational cost. This more or less limits their application. The neural network model developed in this study, on the other hand, provides a framework with a flexible input data type (e.g. 2D land surface observations and others) and complex output datasets (e.g. 3D paleovalley pattern). It is capable to define nonlinear relationships among multiple-support datasets, and employ this relationship for prediction with merely easy-to-obtain input data (now Lines 61-63).

Q2. My trouble with the objectives and problem definition can be best illustrated by first considering the geologic system the authors seek to better map in 3D. 'Paleochannels' can take on a number of different meanings depending on the geologic setting, but from what I can decipher from the introduction, methods and Fig. 2, by 'paleochannels' the authors are referring to incised valley fill deposits like those depicted in Fig. 2b, where the channels are bounded not by adjacent fluvial facies, but by granite. Setting aside for the moment that this looks more like a basin and range style of geologic structure than a paleochannel, based on the vague descriptions in the paper, I can only construe that the flattest portions of the DEM shown in Fig. 2c represent the Quaternary alluvial bottomlands representative of the top of the apparent paleochannels (i.e., top of sed

facies in Fig. 2b). If that is true, the reader's reaction is inevitably: "Why is this even considered a challenging problem? From the topography it is already obvious where these so-called channel deposits locate." Summarizing the case study, it appears that the DEM already nicely identifies locations of the paleochannels, which apparently have been further characterized using AEM, presumably to better identify their depths or depth to bedrock perhaps. This raises the question of what is the problem the authors are attempting to address? If the problem is to better identify x-y locations of the so-called paleochannels, that would appear moot because the DEM already shows them, which also raises the question of why you need DL. If the problem is to better identify paleochannel or incised valley-fill depths, that has apparently already been done with AEM; and furthermore, if the purpose is to use the DL algorithm to map the paleochannels depths so that AEM would not be needed, that also does not appear to make sense because the authors have not established a relationship between the DEM flatness metric and paleochannel depths.

It is possible that if the authors can be more specific about the geology of these 'paleochannel' features that they are trying to map and about what specifically they are trying to accomplish through the application of their ML methods, the above problems would be cleared up. As written, however, the manuscript lacks sufficient definition of the problem, description of their objectives, and description of how their research satisfies those objectives.

Reply: We agree that the problem should be better defined, with greater clarity of objectives and how those were achieved. The following changes have been made in response to the comments.

(1) The Introduction provides background geological information on the palaeovalley system of interest, and why ML is adopted to improve mapping of their location and their 2D/3D geometry (now Lines 74-94).

The case study area is a pre-Pliocene palaeovalley system in central Australia that has

been postulated to contain significant groundwater resources (Dodds and Sampson, 2000). However, their geometry and extent remain largely hidden from view by a valley fill of Pliocene to Pleistocene sediments and overlying Quaternary sand dunes of the Great Victoria Desert (Lewis et al., 2010). Although the thicker valley fill sequences seem to be coincident with contemporary lows or valleys in the more subdued relief of the plains, the definition of the palaeovalley systems remains relatively poor (Munday et al., 2020a). This has been attributed to sandplain sediments forming a relatively continuous cover over much of the Musgrave Province down to 30-40 m depth; below this depth the definition of the palaeovalley systems becomes significantly clearer with a well-defined network of major alluvial channels and tributary systems. As is evident from an analysis of AEM images, the palaeovalley system has a highly irregular geometry with spatially varying depths to basement, and with heterogeneous infill resulting in lithologically controlled palaeovalley aquifers.

Our goal is therefore to develop an efficient and generic machine learning tool to express the relationship between an easy-to-obtain dataset and a more costly dataset for the specific purpose of detecting palaeovalley features that would facilitate the discovery of new groundwater resources in arid and semi-arid regions. In other words, we seek to develop a novel method that uses AEM only for model development on a small training area while the application (i.e. detection of palaeovalleys across large areas) uses readily available landsurface information that otherwise (i.e., without AEM coupling through a training procedure) would have had little value for palaeovalley detection. Moreover, in addition to detection of palaeovalley location, the method should also derive the 3D palaeovalley geometry. Such methodology is premised on the existence of a mechanistic connection between landsurface features and subsurface distribution of palaeovaleys. To what degree such correlation exists (and can be cast in a predictive framework) between palaeovalley geometry and landsurface features derived from digital elevation data in the palaeovalley system of the Musgrave Province will be tested using a deep convolutional neural network methodology.

[Figure]

(2) The paleovalley pattern in this demonstration case is comparable to that of modern valley pattern. Thus, the MrVBF (a 2D land surface observations) is related to the 3D paleovalley structure; but it cannot directly suggest the depth of paleovalley and width of the paleovalley at different depths. AEM-interpreted EC values is a direct index of 3D paleovalley structure (including both depth and width), but it is not available everywhere. We employed our method to define a relationship between MrVBF and AEM-interpreted EC in the data-rich area, and employed in those area where the AEM is not available to predict the 3D paleovalley pattern based merely on the MrVBF (now Lines 167-176).

(3) For the model verification, both the training and validations are conducted in those regions with AEM-interpreted EC. The weights in the neural network model is determined based on the data in the training area. The AEM data in the validation areas is just used to test the ability of the trained model in predicting 3D paleovalley structure, but do not participate in determining the neural network model (now Lines 176-178).

Specific comments in the annotated PDF files

Line 22. Delete 'dramatically'

Reply: Change made.

Line 22-25. "data poor" contradicts "rich/big", and others

Reply: This sentence is now rephrased in Lines 22-24.

Line 29-30. This is most certainly not true of those method, although one might need to use them more expertly (e.g. through zoning of the model region) when non-stationarities are present; Line 32. "is still lacking" to "would be beneficial"; Line 33. "fill this gap". You have not identified as a gap, but rather a potential way of improving upon other methods. Rewrite to better describe the "gap" and what your method potentially does.

Reply: This part is rewritten to present the limitations of existing methods, and the

major problem we wanted to solve with our developed neural network (Line 38-50 and Line 74-94)

Line 39: Add 'e.g.'

Reply: Change made.

Line 119-121. This would appear to represent existing fluvial drainage characteristics. For this to be useful for 'training' the DL model there would have to be a mechanistic connection between these surface features and the subsurface distribution of pale-ochannels. There is a big problem with this approach: modern geomorphic surface characteristics seldom represent or correlate to the morphology and distributions of subsurface facies or rock types.

Reply: Agree.

While the occurrence of palaeovalleys is correlated to the modern-day valley pattern (Jiang et al., 2019), their exact location and geometry in the case study area cannot simply be inferred from modern geometric surface features such as the 2D Multiple-resolution Valley Bottom Flatness (MrVBF) index (calculated from the digital elevation model) (Gallant and Dowling, 2003). The correlation is complicated by the presence of relatively continuous sandplain sediments that cover the palaeovalleys. On the other hand, the vertical structure of a palaeovalley can be interpreted from an airborne electromagnetic (AEM) survey (Ley-Cooper and Munday, 2013; Soerensen et al., 2016). The MrVBF index exists across the entire Australia continent, while AEM data of sufficient spatial granularity only exists in a limited number of prospective mining fields. Our neural network model establishes a relationship between the MrVBF index (high values are indicative of locations with a high probability of deposition of alluvial sediments) and the AEM-interpreted 3D palaeovalley structure. This relationship is then used to predict the 3D palaeovalley structure in those areas with only MrVBF data but without the AEM dataset (now Lines 167-176).

An area 80 km west of the training area is first used to validate the trained neural network in generating 3D PAI. The statistics of squared errors between the simulated 3D PAI and real PAI are calculated at all 200×200×10 voxels. As shown in Fig. 3, the squared error in the training dataset is below 0.1 for 99% of the training domain and with a mean value of about 0.03, and the squared error of the predicted 3D PAI is well below <0.1 for 93% of the validation domain, with a mean squared error of about 0.04. The patterns of the generated palaeovalley in both horizontal and vertical directions align with those inferred from the AEM-derived PAI. This indicates that the deep-learning neural network structure developed in this work is capable of incorporating the relationships between the MrVBF and the buried palaeovalley patterns, and allowing for reliable predictions beyond the training area (Lines 200-207).

Figure 2. No clear. Do the valley bottoms in Fig. 2c correspond each to the type of channel and facies depicted in 2b? If yes, does that mean these are all incised into granite? In that case, the predictive geologic problem would appear to be trivial.

Reply: The valley bottom flatness data from Fig.2c represents the input data for the neural network model, noting that the modern-day valley pattern is correlated with the occurrence of palaeovalleys, however their exact location and geometry in the case study area cannot simply be inferred from the 2D Multiple-resolution Valley Bottom Flatness (MrVBF) index alone. The 2D conceptual model of a palaeovalley (Fig. 2b) is a very simplified representation of the heterogeneous structure of the palaeovalleys in the Musgrave Province. The valley bottoms of Fig 2c have a high likelihood to contain palaeovalley features, incised in a more or less unweathered (resistive) basement rock. This does not make the geologic problem trivial: however, it does provide the basis for delineating the palaeovalley base using a cut-off resistivity boundary. Without such resistivity contrast between basement rock and conductive infill the AEM method would have difficulty in delineating any palaeovalley accurately (now Lines 155-166).

Line 121-123. Is the point here to use AEM results as a ground truth and demonstrate that you could do as good, or almost as good, without the AEM and just using your DL

approach based on surficial information? Not clear.

Reply: Yes. The MrVBF index exists across the entire Australia continent, while AEM data of sufficient spatial granularity only exists in a limited number of prospective mining fields. Our neural network model establishes a relationship between the MrVBF index (high values are indicative of locations with a high probability of deposition of alluvial sediments) and the AEM-interpreted 3D palaeovalley structure. This relationship is then used to predict the 3D palaeovalley structure in those areas with only MrVBF data but without the AEM dataset. For the method verification, both the training and prediction are conducted in the area where AEM data is available. Note that the weights in the neural network are determined based on the training area. The AEM data in the other areas are only used to test the predictive capability of the trained neural network (now Lines 176-178).

**Supplement:**

[revised manuscript text omitted]

---

## Author Response (AR2)

Dear Prof. Wickert,

We greatly appreciate your time in reviewing our article. The comments are constructive and very helpful and improved the quality and hopefully the impact of the paper.

We agree that essential information was missing, especially regarding the genesis, and the resulting complexity, of the palaeovalleys in central Australia. This has been addressed in a new **Section 2.1 Genesis of palaeovalley systems in central Australia**. Next, we appreciate your comment about generalizability of our methodology, specifically about the lack of information that allows the reader to make their own determination about the potential usefulness of this methodology for their landscape/application. This has been addressed by yet another new **Section 4.3 Generalisation**. We trust that these major modifications have made the manuscript acceptable.

Other minor comments have been addressed as well. The detailed responses are listed below, with the changes marked in the annotated manuscript.

We look forward to discuss with you any further queries.

Thanks again.

Best Wishes,

Zhenjiao

**Responses to editor**

**Generalizability.** Your study site has been quite tectonically stable for a very long time, and was never glaciated. Therefore, the geomorphic evolution is straightforward: erosoin of bedrock uplands produces sediments that accumulate in basins, and geological and climatic processes lead to the formation and filling of valleys. In this case, these valleys have been loci of deposition throughout the Cenozoic. However, Reviewer 2's comment notes that your situation does not match that of many other parts of the world, where surface topography and subsurface structure are not linked. As an example that I know well, note Minnesota (USA)'s depth to bedrock (https://mngs-umn.opendata.arcgis.com/}, topography (http://arcgis.dnr.state.mn.us/maps/mntopo/), and aeromagnetic anomalies (https://mngs-umn.opendata.arcgis.com/app/the-aeromagnetic-database). You would not obtain the same answer with your methods here, where I am! However, as you note, it works for both your training and test regions, both of which are within the same physiographic and geological region. Therefore, I think that you need to couch your algorithm as an approach to find and characterize valley fill when topography is still present. And in so doing, it might be good to look into the work of Mey et al. (2015: {https://agupubs.onlinelibrary.wiley.com/doi/full/10.1002/2014JF003270}, who also address this

**Reply:** A new section **4.3 Generalisation** has been introduced to address the above comments (**Line 356-415**). We acknowledge that there was insufficient description of the site features for readers to assess to what degree the methodology could be transferred to other regions. We trust this has now been addressed, giving the reader guidance about topographical and other landscape features that allowed a successful application of proposed model using topographic predictors such as MrVBF.

**4.3 Generalisation**

[revised manuscript text omitted]

In addition, the Mey et al. (2015) reference was cited in **lines 50-52**. For clarification, the trained neural network in this study is merely workable for palaeovalley prediction in central Australia, i.e. the underlying relationship in the trained model should be consistent to those regions for which prediction are made. For the application in other regions and other subsurface structures of interest, the model will need to be re-trained based on the specific input and output datasets. Alternatively, other predictors may be used, such as remote sensing data. In this regard **Figure 2** is also modified to make the input data more general.

**Fitting geological reality or a geophysical inversion?** I am concerned about your note regarding the valley structure being complex and discontinuous. This is inconsistent with your description of fluvial deposits, which should form a continuous sedimentary package (i.e., streams flow consistently downhill). There are two options. One is tectonics and localized deformation that reshaped the contact between the basement rock and the overlying fluvial deposits. The other option, which I find to be more compelling, is that the geophysical inversions may be imperfect. Although I am not as familiar with aeromagnetic data, I believe that inverse-distance effects would cause narrow sections of the paleovalley to be harder to detect and/or to seem to terminate at a shallower depth than wider portions of the valley. If I am correct, then your ML-based fit would be tuned to the details in a depth-and-wavelength-sensitive geophysical inversion rather than designed to match geological reality based on lithological data and interpretations.

**Reply:** We included a new section (**Section 2.1**) describing the genesis of the palaeovalleys in central Australia. This will illustrate their complexity, including the role of tectonics and other processes. We also argue that at least for the larger structures, the model fit is primarily using an accurate representation of geological reality, recognizing that by its nature the geophysical inversion will always be a simplification of true geology.

**2.1 Genesis of palaeovalley systems in central Australia**

[revised manuscript text omitted]

We further made the following changes to support the description of the palaeovalley structure:

(1) **Figure 2b** is redraw to better describe the general lithofacies in the palaeovalley in this study area. The lithofacies feature coarse sands deposits gradually evolving to the clay-dominant depositions, overlying by the aeolian sands and silts. This represents the depositional environment changing from wet

to dry periods. The sudden lithofacies change in the previous image was quite misleading.

(2) It was found that the high bulk electrical conductivity values (EC) are a proxy for palaeovalley presence in contrast to the low EC of the bedrock; the higher EC, the higher probability of the palaeovalley presence (**lines 254-257**). We thus use an AEM-derived index to indicate the occurrence of palaeovalley. We also clarify that the model fit is based on the geophysical inversion rather than the lithological data (**lines 261-264**).

**Line 12.** tomography → topography?

**Reply:** This is now modified in **line 13**.

**Line 75**. their → its

**Reply:** This is now corrected in **line 20.**

**Line 79.** "sandplain" is not a genetic term. Later in the paper, it seems that you indicate this to be aeolian. You should make this clear here, because it importantly indicates that you are inverting across a buried and preserved paleolandscape and its deposits. (You can expand upon this to in regards to the geological setting.)

**Reply:** The 'sandplain' is now replaced by 'aeolian' for clarification.

**Line 82.** "As is evident from an analysis of AEM images, the palaeovalley system has a highly irregular geometry with spatially varying depths to basement, and with heterogeneous infill resulting in lithologically controlled palaeovalley aquifers". This is the core of my second major point, above.

**Reply:** This has been addressed in the **new Section 2.1**.

**Line 85-87.** "Our goal is therefore to develop an efficient and generic machine learning tool to express the relationship between an easy-to-obtain dataset and a more costly dataset for the specific purpose of detecting palaeovalley features that would facilitate the discovery of new groundwater resources in arid and semi-arid regions." See the first point; I think that this might not be as generally applicable as you state, especially since you test and validate it in the same stable geological setting.

**Reply:** We agree that this statement was somewhat optimistic in regard to the success of DEM data as predictor variable, while the methodology itself is still sufficiently generic such that other data types can be used for prediction purposes. Text has been rephrased to emphasize that we are using this study area as an example to demonstrate the effectiveness of the method. For the application in other areas with other subsurface structures of interest, the model will need to trained again based on the problem and data at hand (**e.g. Line 56-59, Line 69-71 and Figure 2**). Also, while other predictor data may be more suitable than DEM data, the method itself is sufficiently generic that it can be easily adopted.

**Line 90-91.** "Such methodology is premised on the existence of a mechanistic connection between landsurface features and subsurface distribution of palaeovaleys". This indeed is THE key limitation, and I think it deserves to be more clearly stated, as this will help readers at once recognize whether your

approach is one that they could use in their system or not.

**Reply:** Thank you, good point indeed. The new **Section 4.3 Generalizability** addresses this in detail.

**Line 103,** Fig. 1, Is your use of "image" standard? Because to me, these are "arrays" or "layers of inversions", but not actually images.

**Reply:** Following the terminology in deep learning, we use the 'image' to express the input and output data. Considering this comment, the '3D array' is annotated following the first appearance of the 3D image in **Figure 2 and line 184**, for clarification.

**Line 154.** This is your first mention of a CSIRO data set. What is it?

**Reply:** This is now explained in **line 253** and the link is given in the **code/data availability section.**

**Line 157.** I assume that this is the ``sandplain", per my above comment.

**Reply:** This is now replaced or clarified with 'aeolian'.

**Line 159.** The MODERN valley bottom. In addition, you have not yet introduced the valley-bottom flatness index. It seems that this paragraph may be out of place, and should follow the paragraph below in order to provide the requisite information first. **Line 169.** Define how the MrVBF is calculated and the details of the source DEM (which data source, resolution, etc.). **Line 176.** MrVBF is a derived data product, not a data source in itself.

**Reply:** This part is now rewritten in **lines 218-254.**

**Line 183.** Not every deposit filling a valley is an aquifer. Is this formula sensitive to the nature of the sedimentary fill? Associated with that, the cross section in Fig. 2B indicates an aquitard between two aquifers.

**Reply:** Yes, the previous figure 2b is quite misleading, we now redraw **Figure 2b** to better express the general lithofacies in the palaeovalley.

**Line 228.** What is a fully connected layer in the encoder? Could you provide a bit more ML background to help the reader to understand why it is important and creating the observed result?

**Reply:** This is now explained in **lines 307-308.**

**Line 287-289.** I appreciate your desire to contribute, but do not think that your tool has generic applicability based on the points that I have raised.

**Reply:** This sentence is now rephrased in **lines 426-428.** We trust that the major changes made throughout the manuscript will make clear what landscape features are required for this DEM-based method to be have more general applicability.

---

## Author Response (AR3)

l 91-92: "It is also worthy of note that the topography of the study area is very subdued, with contemporary draining systems being discordant with respect to their ancient precursors.": I had thought that the key to your work was the fact that the topoography, while subdued, was concordant with the paleovalley systems.

Our modified text is: *It is worthy of note that the topography of the study area is very subdued, and whilst the contemporary draining channels are discordant with respect to their ancient precursors as defined by the thalweg or deepest part of the palaeovalley, these old valley systems are concordant with the subdued valley forms expressed in today's landscape.*

l 98: "Ma" alone -- "ago" not needed.

OK, remove the ago.

l 108: The existence of plays may bolster your counterargument to my last review: either sediment deposition or neotectonics must have split the valley into internally drained basins by this time.

Our modified text is as follows: *During the Late Miocene to Early Pliocene (about 10-3 Ma ago), evaporation of these sediments led to the deposition of a gypsum layer which was accompanied by intermittent fluvial deposition. A combination of active faulting and sedimentation may have encouraged the development of small, narrow internally draining basins during this period.*

l 135-139: You state that the geophysical data match geological reality, but what would really be good is to cite sources (e.g., with borehole logs) to indicate this.

Our modified text is as follows text: The *geophysical expression is well matched with the geological reality in that targeted drilling (described in Krapf et al 2019, and Munday et al. 2020), confirmed the presence of thick (>150m) alluvial sediment fill sequences associated with the interpreted palaeovalleys, which were also coincident with the more conductive linear features identified in the AEM data.*

l 252: I would suggest formatting this as a full reference to go in the reference list, especially considering that it is a doi. If it stays a link, the Copernicus staff will ask you to give the date of last access.

Cite as: Munday, Tim (2019): Musgrave Province Airborne Electromagnetic Conductivity Grids. v1. CSIRO. Data Collection. https://doi.org/10.25919/5d0868d48591e

l 384-385: Not sure if this sentence is needed; the prior sentence states the same, but more precisely.

OK – Remove this sentence: However, the geologic/geomorphologic complexity of the Australian palaeovalley systems is therefore no less.